# Post-Transcriptional Regulatory Crosstalk between MicroRNAs and Canonical TGF-β/BMP Signalling Cascades on Osteoblast Lineage: A Comprehensive Review

**DOI:** 10.3390/ijms24076423

**Published:** 2023-03-29

**Authors:** Hui-Yi Loh, Brendan P. Norman, Kok-Song Lai, Wan-Hee Cheng, Nik Mohd Afizan Nik Abd. Rahman, Noorjahan Banu Mohamed Alitheen, Mohd Azuraidi Osman

**Affiliations:** 1Department of Cell and Molecular Biology, Faculty of Biotechnology and Biomolecular Sciences, Universiti Putra Malaysia, Serdang 43400, Selangor, Malaysia; 2Department of Musculoskeletal and Ageing Science, Institute of Life Course and Medical Sciences, University of Liverpool, Liverpool L7 8TX, UK; 3Health Sciences Division, Abu Dhabi Women’s College, Higher Colleges of Technology, Abu Dhabi 41012, United Arab Emirates; 4Faculty of Health and Life Sciences, INTI International University, Persiaran Perdana BBN, Putra Nilai, Nilai 71800, Negeri Sembilan, Malaysia

**Keywords:** microRNAs (miRNAs), bone, osteoblast lineage, TGF-βs, BMPs, Smads, osteogenic differentiation, bone diseases

## Abstract

MicroRNAs (miRNAs) are a family of small, single-stranded, and non-protein coding RNAs about 19 to 22 nucleotides in length, that have been reported to have important roles in the control of bone development. MiRNAs have a strong influence on osteoblast differentiation through stages of lineage commitment and maturation, as well as via controlling the activities of osteogenic signal transduction pathways. Generally, miRNAs may modulate cell stemness, proliferation, differentiation, and apoptosis by binding the 3′-untranslated regions (3′-UTRs) of the target genes, which then can subsequently undergo messenger RNA (mRNA) degradation or protein translational repression. MiRNAs manage the gene expression in osteogenic differentiation by regulating multiple signalling cascades and essential transcription factors, including the transforming growth factor-beta (TGF-β)/bone morphogenic protein (BMP), Wingless/Int-1(Wnt)/β-catenin, Notch, and Hedgehog signalling pathways; the Runt-related transcription factor 2 (RUNX2); and osterix (Osx). This shows that miRNAs are essential in regulating diverse osteoblast cell functions. TGF-βs and BMPs transduce signals and exert diverse functions in osteoblastogenesis, skeletal development and bone formation, bone homeostasis, and diseases. Herein, we highlighted the current state of in vitro and in vivo research describing miRNA regulation on the canonical TGF-β/BMP signalling, their effects on osteoblast linage, and understand their mechanism of action for the development of possible therapeutics. In this review, particular attention and comprehensive database searches are focused on related works published between the years 2000 to 2022, using the resources from PubMed, Google Scholar, Scopus, and Web of Science.

## 1. Introduction

Bone is the dynamic organ that constitutes the internal framework or foundation of the human body [1]. The skeleton makes up 20% of body weight and performs various functions including providing structural support for the rest of the body, permitting movement and locomotion, protecting vital internal organs and structures, maintaining mineral homeostasis and acid-base balance, serving as a reservoir of growth factors and cytokines, and contributing the environment for haematopoiesis within the marrow spaces [1,2]. The bone is constructed from three crucial processes: bone ossification, modelling, and remodelling [2,3,4,5].

The word “osteoblast” is derived from Greek, by which “osteo” means bone, while the suffix “blast” means to germinate. Ossification or osteogenesis is the process of bone formation by osteoblasts [5]. Skeletal development is a multistep process during which mesenchymal progenitor cells (MSCs) undergo proliferation and differentiation, giving rise to cartilage and bone cells [6,7]. There are two types of osteogenesis pathways; intramembranous and endochondral ossification, which arise from two different embryonic populations of MSCs [8]. Both intramembranous and endochondral ossification processes commence with osteoprogenitor cells (or pre-osteoblasts) [8]. Osteoblasts are then produced from the osteoprogenitor cells by various regulatory and signalling cascades [8]. Osteoblasts that are derived from neural cells of neural ectoderm are directly formed from condensed mesenchymal progenitors without an intermediate chondrogenic stage [9,10]. These osteoblasts form the flat bones of the face, most of the cranial bones, and the clavicles (collarbones) via intramembranous ossification [11]. The embryonic origin of the axial skeleton and appendicular skeleton osteoblasts are derived from lateral plate mesoderm and paraxial mesoderm, respectively [12]. Both skeletal tissues are built from endochondral ossification which requires the formation of cartilaginous templates that prefigure future skeletal elements and are later replaced by mineralized bone [10,11].

The dynamics or plasticity of bone is achieved through the process of bone modelling and remodelling throughout life which gives rise to the overall shape, size, and strength of the bone [13]. Bone modelling normally occurs during childhood, with changes to the bone’s overall shape and size in accordance with the physiological factors and mechanical forces. This leads to gradual adaptation of the encountering forces by the skeleton [2,14]. Bone remodelling is part of a lifelong process of bone turnover which involves sequential steps through which old, micro-damaged bone (bone resorption) is replaced with new mechanically stronger, and healthier bone (bone formation) [2,14,15,16]. Bone remodelling is required for the preservation of bone mechanical strength and integrity, and the maintenance of normal bone homeostasis [14].

Both essential events of bone homeostasis are through the activities of bone-forming osteoblasts, which arise from mesenchymal stem cells (MSCs), and the bone-resorbing osteoclasts, which originate from haematopoietic stem cells (HSCs) [4,15,17]. The process of bone resorption and bone formation is not tightly regulated in bone modelling, as the osteoblasts and osteoclasts act independently towards biomechanical forces at distant skeletal sites or distinct anatomical locations [10,18]. In contrast, bone remodelling required tightly coupled action of both bone resorption and bone formation. This process occurred at the bone remodelling site, organized in specialized units called bone multicellular units (BMU) [10,15,19,20]. The chronologically and quantitatively balanced actions of osteoblast and osteoclast lineage cells are stringently regulated by both local and systemic factors and key signalling pathways [10,15,19,20]. Any significant alterations to the progress of osteoprogenitor cell commitment and osteogenic differentiation, and/or deviations from a neutral balance between resorption and formation would mean severe accelerated bone loss or bone gains [15]. These situations will possibly lead to disastrous consequences and impairments in terms of abnormal skeletal development, increased fracture risk or compression syndromes [10,15].

Recently, there has been an increasing number of excellent reviews that have documented the characterization of a class of small regulatory molecules that modulating distinct aspects of bone development and formation. These small molecules were noted to post-transcriptionally regulate a number of critical osteogenic signalling pathways, including the wingless-type (Wnt), transforming growth factor-beta (TGF-β), and bone morphogenetic protein (BMP) pathways. One such molecule with a major impact on bone is known as microRNA [20,21,22].

MicroRNAs (miRNAs) are a family of small, endogenous, single-stranded, and non-protein coding RNAs about 19 to 24 nucleotides in length, which are evolutionarily conserved across species [23]. MiRNAs play a key regulatory role in gene expression at the post-transcriptional level in various biological processes and developmental stages via extremely specific and complex regulatory networks.

Many miRNAs are involved in the regulation of various crucial biological processes such as cell proliferation, differentiation, cell cycle regulation, apoptosis, cellular metabolism, homeostasis, and immune response [21,24,25,26]. Aberrant expression or dysregulation of a single or small subset of miRNAs can led to intense cellular outcomes or disease developments including cancers, cardiac failures, viral diseases, neurodevelopmental disorders, immune-related diseases, musculoskeletal, and bone-related diseases [23,27,28].

Our present review integrates current information on how the miRNAs can positively and negatively regulate osteogenic differentiation and functions by supervising the canonical TGF-β/BMP signalling pathways in osteoblastic lineage. TGF-βs and BMPs orchestrate diverse functions in skeletal development and formation, including mesenchyme condensation, skeleton morphogenesis, growth plate development, and on osteoblast commitment, differentiation, and activity. In addition of discussing the contribution of these miRNAs towards osteoblast differentiation and bone development through the modulation of TGF-β and BMP signalling pathways, their potential applications as therapeutic targets are also emphasized.

## 2. MicroRNAs: A Small Molecule with Great Regulatory Functions

In 1993, the Lee and Wightman groups discovered the first miRNA, *lin-4* as a small RNA transcribed by the *lin-4* locus of *Caenorhabditis elegans* (*C. elegans*) nematodes [29,30]. Since its discovery, miRNAs are found to be present across the animal, plant, and other eukaryotic systems [31]. MiRNA profiling strategies such as qRT-PCR, microarray, and RNA sequencing have dramatically increased the amount of miRNAs data [32,33]. Additionally, computational algorithms or bioinformatics approaches had been developed for the systemic management or organization of these data to ease the retrieval and distribution of information [34]. Thus far, the latest miRbase miRNA sequence database (Release 22.0, March 2018, http://www.mirbase.org/) (accessed on 20 October 2021) had deposited over 38,589 hairpin precursors and 48,860 mature miRNAs that were distributed over 271 species [35]. There are about 2600 mature microRNAs that are encoded in the human genome according to miRBase v.22 [36]. Additionally, it is estimated that about 60% of all mammalian protein-coding genes contained at least one miRNA regulatory binding site [37].

In mammals, the first step of miRNA biosynthesis begins with the transcription of miRNA genes in the nucleus by RNA Polymerase II or III to generate the primary transcripts (pri-miRNAs) [38,39,40,41]. The pri-miRNAs which are several kilobases in length with local stem-loop structures, 5′-cap and poly(A) tail [38] are then processed into the precursor miRNAs (pre-miRNAs) by the microprocessor complex, composed of an RNA binding protein DiGeorge Syndrome Critical Region 8 (DGCR8) and Drosha (a ribonuclease III enzyme) [42,43,44,45,46]. The pre-miRNAs formed are 70- to 80-nucleotides long hairpin-shaped structure, with a 5′-phosphate and two-nucleotides 3′-overhang [47]. Next, the pre-miRNAs are exported from the nucleus into the cytoplasm by exportin-5 (XPO5) that belongs to the RanGTP-dependent nuclear transport reporter family [48,49,50,51].

In the cytoplasm, pre-miRNA has undergone cleavage near the terminal loop by another RNase III enzyme known as Dicer [52,53]. The activity of Dicer is aided by the cofactor protein called transactivation response RNA binding protein (TRBP) to generate a 20-nucleotides long mature miRNA/miRNA* duplex [53,54,55,56,57]. The miRNA/miRNA* duplex is then loaded into a member of the Argonaute (Ago) protein subfamily and resulted in the formation of RNA-induced silencing complex (RISC) [58,59]. Mostly, the passenger strand (miRNA* strand) is degraded whereas the guide strand (mature miRNA strand) remains bound to the RISC [60]. The Ago protein-bound mature miRNA is subsequently assembled into an effector complex known as the miRNA-containing RNA-induced silencing complex (miRISC) [61]. Within the miRISC, the miRNA “seed sequence” with 2- to 8-nucleotides at the 5′-end will predominantly undergo base-pairing with the mRNA 3′-untranslated region (3′-UTR) [62,63]. However, some reports stated that miRNAs will also bind to the 5′-untranslated region (5′-UTR) and open reading frame (ORF) [64,65]. Figure 1 illustrates the pathway of miRNA biogenesis and miRNA activity.

MiRNAs post-transcriptionally regulate gene silencing either through mRNA cleavage or translation repression, based upon the degree of sequence complementarity or homology between the miRNA and mRNA. Perfect complementation will result in mRNA degradation or cleavage, while imperfect complementation gives rise to translation inhibition or repression [40].

Owing to the high degree of sequence complementarity between seed and target sequences, a single miRNA can recognize multiple mRNA transcripts and modulate their functions [36]. At the same time, a single mRNA molecule typically has multiple miRNA binding sites that can be cooperatively targeted by a vast number of distinct miRNAs [36]. miRNAs also work in a concerted action along with other classes of non-coding RNAs, such as long non-coding RNAs (lncRNAs) and circular RNAs (circRNAs), when modulating the mRNA functionality [66].

Similar to miRNAs; lncRNAs (with a length exceeds 200 nucleotides) and circRNAs (composed of closed-loop structure, without terminal 5′-caps and 3′-polyadenylated tails) are members of the non-coding RNAs (ncRNAs) [67]. LncRNAs and circRNAs may act as competing endogenous RNAs (ceRNAs) or work as the “miRNA sponges” by competitively binding to miRNAs’ response elements, thus sequestering the miRNAs from their downstream target mRNAs. This results in indirect regulation on the expression levels of the miRNAs’ target mRNAs, with either positive or negative effects on biological and pathological processes [67].

## 3. The Osteogenic Regulating Actions of MicroRNAs and Protein Catalysts Involved in MicroRNAs’ Biogenesis

The mechanism involved in the differentiation of MSCs to osteoblasts is a complicated process that encompasses three potential stages: (a) cell proliferation, (b) cell differentiation while maturing the extracellular matrix, and (c) matrix mineralization [8]. A comprehensive understanding of the relationships behind miRNAs’ mode of action, miRNA biosynthesis machinery, and the mechanism of osteoblast differentiation and functions is important for the development of new therapeutic strategies. This may lead to the progress of clinical applications in enhancing bone formation and treating systemic bone diseases such as osteoporosis.

Osteoporosis is the common bone disorder characterized by low bone mass, decreased bone strength, and micro-architectural deterioration of bone tissue, which is highly associated with disrupted balance of bone remodelling [15]. This “porous bone” condition resultant in bone fragility and increased risk for sudden and unexpected bone fractures. Osteoporosis is traditionally grouped into the primary and secondary osteoporosis [15]. Primary osteoporosis is caused by natural age-related changes to bone density and included both postmenopausal (Type I) and age-related or senile osteoporosis (Type II) [10,15]. Secondary osteoporosis may arise from secondary complication of certain underlying diseases or medical conditions (e.g., endocrine disorders, genetic diseases, gastrointestinal and nutritional disorders, connective tissue disorders, metabolic disorders, and haematological disorders), consequences of changes in physical activity (e.g., immobilization), or adverse results of therapeutic interventions or medications (e.g., glucocorticoid) [10,68].

Anti-osteoporotic treatments can be divided into two groups: (a) anti-resorptive therapies that prevent bone resorption by interfering with osteoclasts’ biological function; and (b) anabolic therapies that encourage bone formation by facilitating the rate of bone remodelling [69]. Although these anti-osteoporotic medications can have a remarkable impact on osteoporosis, their long-term use is constrained by possible side effects (e.g., gastrointestinal intolerance, osteonecrosis, excessively suppressed bone turnover, thromboembolic disease, and increased risk of cancers including osteosarcoma, ovarian/endometrial/breast cancers) [69]. Considering the above matters, there is still a high demand for the invention of a novel and more effective anti-osteoporotic medication that have a wider therapeutic spectrum with fewer side effects. It has been greatly discussed throughout this review that miRNAs have shown remarkable therapeutic potential for osteoporosis and associated fractures, owing to their post-transcriptional regulatory roles on the metabolism, proliferation, and differentiation of bone cells. Therefore, miRNA-based therapeutic approaches, either through gain-of-function therapies that restore the expression of suppressed miRNA in disease conditions, or loss-of-function therapies that attenuate the activity of overexpressed miRNA during the pathological process, can shed a light as future anti-osteoporotic measures in clinical practice [69].

In a study conducted by Sun, Zhongyang et al. high expression of miR-103-3p was found to be resulted in reduced bone formation capacity in bone specimens isolated from elderly women with fractures, and ovariectomized (OVX) mice. MiR-103-3p directly targeted methyltransferase-like 14 (*Mettl14*) to inhibit osteoblast activity in terms of cell proliferation, differentiation, and matrix mineralization in vitro, and inhibit bone formation in vivo. It holds one’s attention that, METTL14-dependent N^6^-methyladenosine (m^6^A) methylation was found to provide negative feedback action on miR-103-3p’s recognition and processing by the microprocessor protein DGCR8, and subsequently promoted the osteoblast activity. The miR-103-3p/METTL14/m^6^A signalling axis may therefore serve as a potential target for ameliorating the bone mass deterioration in postmenopausal osteoporosis [70].

Interestingly, conditional knockout of *DGCR8* in osteoprogenitor cells of mice increased bone volume (BV/TV), trabecular number (Tb/N), and trabecular thickness (Tb.Th), but decreased trabecular separation (Tb.Sp) in femurs derived from mice aged 5-, 7-, 12-, 24-, and 36-weeks. Von Kossa, and tartrate-resistant acid phosphatase (TRAP) staining, in addition to calcein double labelling also revealed that osteoblast formation was increased in *Dgcr8* conditionally knockout mice. It was further identified that *osteocalcin* (*OCN*) transcript elevated during the differentiation event of primary osteoblasts isolated from the calvaria and long bones of the *Dgcr8*-deficient new-born mice. Next, miR-22 that potentially targeted *OCN* was elucidated to be downregulated in the *Dgcr8* conditionally knockout mice, thereby may promote osteoblast differentiation in a DGCR8-dependent manner [71].

Shirazi, Sajjad et al. demonstrated that mRNA and protein knockdown of Dicer and Argonaute 2 (Ago2) in primary human bone marrow-derived mesenchymal stem cells (BMSCs) may result in global reduction of miRNA biosynthesis and impairment of BMSCs’ responses to miRNAs. Knockdown of both Dicer and Ago2 also suppressed osteoblastic differentiation of human BMSCs, as evidenced by a reduction in culture mineralization and osteoblastic gene expression. Additionally, the knockdown of Dicer led to a reduction of the quantity and the range of miRNAs expressed in human BMSC exosomal cargos. MiR-183-5p and miR-411-5p were among the top 25 downregulated miRNAs in Dicer-deficient human BMSCs, while miR-515-5p, miR-151a-3p, miR-103a-3p, miR-191-5p, miR-320a-3p, and miR-6894-3p were the six upregulated miRNAs. Further pathway analysis revealed that the TGF-β, Hippo, and extracellular matrix (ECM) signalling pathways may be controlled by these dysregulated miRNAs. Moreover, treatment of the Dicer-deficient human BMSCs with the wild-type MSC-derived exosomes and mimics of the downregulated miRNAs could successfully recover the functions of exosome-mediated signalling, and effectively salvage the impaired osteogenic differentiation [72].

On the other hand, depletion of the key miRNA-processing enzyme, Dicer by apoptosis inducers or siRNA silencing resulted in increased apoptosis in mice osteoblasts. Oestrogen protects the adult skeleton against bone loss by mediating anti-apoptotic effects on the osteoblasts. The expression of Dicer is positively regulated by oestrogen. Furthermore, endogenous oestrogen depletion via ovariectomy procedure in mice resulted in a noticeable decline in bone mineral density (BMD) and trabecular number. Additionally, decreased expression of Dicer and miR-17-92a, and increased expression of BH3-only protein Bim were observed in osteoblasts derived from the trabecular bone of OVX mice, which executed a highly apoptotic function [73].

The differentiation and fate of osteoblasts are also modulated by a plethora of extracellular signals, such as transforming growth factor-beta (TGF-β)/bone morphogenic protein (BMP), Wingless/Int-1(Wnt)/β-catenin, Notch, and Hedgehog, to facilitate the skeletal development [8,74]. Over the last decade, these signalling pathways have also been shown to direct the expression and activity of osteogenesis-related transcription factors such as runt-related transcription factor 2 (RUNX2) [75], and osterix (Osx) [76]. Under physiological conditions or even in pathological scenarios, the selective expression of miRNAs that promote or repress the osteogenesis is closely linked to the sequential signalling pathways and transcription factors involved in the different steps of osteoblasts differentiation. MiRNAs are thus essential in a lot of biological feedback loops, including in the regulation of osteoblast biology. The following section discusses on how the miRNAs involved in osteoblast differentiation via TGF-/BMP signalling pathway modulation, in recently reported in vitro and in vivo experiments.

## 4. Transforming Growth Factor-Beta (TGF-Β)/Bone Morphogenic Protein (BMP) Signalling

Transforming growth factor-β (TGF-β) superfamily is composed of more than 40 members of ligands, including TGF-βs, bone morphogenic proteins (BMPs), growth differentiation factors (GDFs), nodal-related proteins, and activins [77,78]. These members have extensively exerted pleiotropic actions on a multitude of cellular processes, such as gene expression, cell growth, cell survival, cell migration, cell fate decision, and cell lineage proliferation and differentiation, both during embryogenesis and in the homeostasis program of adult tissues [78,79].

TGF-β superfamily ligands can initiate signalling cascades via both canonical Smad-dependent and non-canonical Smad-independent pathways [80]. In canonical signalling, the name ‘small mothers against decapentaplegic’, or ‘Smad’ for short, was derived from the Sma and Mad proteins of *Caenorhabditis elegans* and *Drosophila*, respectively [81]. Three types of Smads have been described in mammals according to their functions: the receptor-regulated Smads (R-Smads), common-partner or comediator Smads (Co-Smads), and inhibitory Smads (I-Smads) [81,82,83].

Canonical signalling is initiated upon the binding of TGF-β or BMP ligands to a heterotetrametric receptor complex in the cell membrane [84,85,86]. This heterotetrametric receptor complex is composed of type I and II serine/threonine kinase receptor [84,85,86]. Upon binding of ligands, the type II receptor will trans-phosphorylate the corresponding type I receptor, which then recruits and phosphorylates their specific receptor-Smads (R-Smads) [84,85,86]. The phosphorylated R-Smads associate with the common-partner Smad (Co-Smad) to form a heterocomplex. Consequently, the R-Smad/Co-Smad complex translocates into the nucleus, where it binds to promoters of target genes and governs their transcription, in cooperation with other DNA-binding transcription factors, co-factors, co-repressors, and chromatin remodelling factors [81,82,83]. In addition, TGF-β/BMP signalling can also be regulated by multiple Smad-independent signalling pathways, including extracellular signal-regulated kinase (ERK), c-Jun N-terminal kinase (JNK), p38 mitogen-activated protein kinases (p38 MAPK), Ras homolog family member A (RhoA), phosphatidylinositol 3-kinase (PI3K), and protein kinase B (Akt) in a cell-context-dependent manner [87].

TGF-β and BMP signalling pathways are antagonized by different components. These antagonists are comprised of the extracellular BMP-binding proteins, such as Noggin, Chordin, Gremlin, and Follistatin; the membrane-pseudo receptors, such as Bambi and Crim1; and the intracellular inhibitory Smad proteins, such as Smad6 and Smad7 [88,89,90]. In addition, the binding of latent TGF-β binding protein (LTBP) to TGF-β can affect the bioavailability of TGF-β, and thereby block the interaction between TGF-β and its receptors [91]. The following section provides a description on miRNA’ involvement in osteogenic differentiation and how these small molecules regulate genes implicated in the canonical TGF-β, BMP, and Smad signalling cascade. Figure 2 illustrates the signalling components involved in the regulatory pathway of TGF-β/BMP/Smad network.

## 5. MicroRNAs Regulating the Transforming Growth Factor-β (TGF-β) Signalling Pathway

### 5.1. MicroRNAs and Transforming Growth Factor-β (TGF-β) Ligands

Transforming Growth Factor-β (TGF-β) signalling has been proved to established pleiotropic involvement in osteoblast differentiation [92,93]. Moreover, TGF-β/Smad signalling is responsible for the regulation of mesenchymal lineage commitment and osteoblastic differentiation via a stage-specific manner. TGF-β signalling promoting the recruitment, enhancing proliferation, and providing competence for early-stage differentiation of osteoprogenitor cells [92,94]. However, TGF-β signalling inhibits terminal-stage osteoblast differentiation and matrix mineralization [92,94]. TGF-β has three isoforms in mammals: TGF-β1, TGF-β2, and TGF-β3 [77,95]. *TGF-β1* deficient mice showed a remarkable reduction in bone growth and mineralization [96]. A gain-of-function mutant of *TGF-β1* resulted in Camurati-Engelmann disease (CED, *OMIM#13100*) in humans, a form of skeletal condition characterized by abnormally thick bones (hyperostosis) primarily affecting the long bones of the arms, legs, and skull [97]. Additionally, *TGF-β2* and *TGF-β3* double knockout mice had severe malformations in rib and sternum development, failure of anterior body wall fusion, and experienced early embryonic lethality [98].

It was demonstrated that the TGF-β produced by osteoblasts could be stored in a latent form in the bone matrix [99]. TGF-β has been shown to stimulate collagen synthesis, and it is an effective chemokine that increases the extracellular matrix and regulates bone and cartilage formation, which could thereby serve as a crucial local and systemic regulatory molecule involved in fracture healing [99,100,101,102]. There are indications from studies in bone fracture healing that miR-185 is a key regulator of TGF-β1. Zhao, Pengfei et al. found an inverse relationship between TGF-β1 and miR-185 expression in the bone tissue, blood, and cerebrospinal fluid of patients with spinal cord injuries induced by thoracolumbar spine compression fractures. Expression levels of TGF-β1 mRNA and protein were significantly increased in patients who underwent surgery 8–14 days following fracture compared to those that who underwent surgery 1–7 days following a fracture. In addition, miR-185 expression was significantly decreased following the time of injury. These findings suggest that miR-185 target *TGF-β1*, and that both molecules could serve as important diagnostic tools in monitoring the status of skeletal repair and regeneration [103]. Sun, Deping et al. also observed that TGF-β1 expression was gradually elevated in the bone and blood tissues within two weeks post ankle fracture, accompanied by downregulation of miR-185 expression. The same study found that upregulation of miR-185 could reduce the proliferative activity of hFOB1.19 cells via influencing *TGF-β1* gene transcription and translation [104].

Osteonecrosis of the femoral head (ONFH) is an orthopaedic disease caused by compromised blood supply of the femoral head, leading to local death of the osteocytes and the component of bone marrow [105,106]. Subsequently, this condition led progressively to advanced structural deterioration and collapse of the femoral head, and ultimately caused pain, dysfunction of the hip joint and severe arthritis [105,106]. Tian, Lei et al. reported increased expression of miR-141 and decreased expression of TGF-β2 in femoral head tissues of ONFH patients. Suppression of miR-141 resulted in an elevated level of TGF-β2, by which *TGF-β2* is a direct target of miR-141. Furthermore, the suppression of miR-141 or elevation of TGF-β2 inhibited apoptosis of bone cells derived from ONFH rats, elevated osteoprotegerin (OPG), B-cell lymphoma 2 (Bcl-2), BMP-2, and RUNX2 expression for stimulating osteoblast activity, with concomitant decreases in osteoprotegerin ligand (OPGL), Bcl-2-associated X protein (Bax), and receptor activator of nuclear factor kappa-β (RANK) expression for inhibiting osteoclast activity in the femoral head tissues of rats with ONFH [107].

MiR-140-5p and miR-140-3p expression was found to be enriched in undifferentiated human MSCs, thought to support cell proliferation and stemness. Expression of these miRNAs decreased during MSC maturation then increased again in the later phase of osteogenic differentiation [108]. Fushimi, Shigeko et al. also showed that miR-140-3p expression decreased in response to the overexpression of Wnt family member 3A (*Wnt3a*) in osteoblastic MC3T3-E1 cells. *TGF-β3*, which was shown as a direct target of miR-140-3p was increased in response to Wnt3a overexpression. Transfection of miR-140-3p mimic into MC3T3-E1 cells resulted in inhibition of TGF-β3 expression and transcriptional activation of late osteoblast marker gene *OCN*. These studies therefore suggest existence of a regulatory relationship between Wnt3a and TGF-β3 signalling pathways in MC3T3-E1 osteoblastic differentiation, with miR-140-3p as a key regulatory factor [109].

Huang, Yiping et al. highlighted that lncRNA-H19 and miR-675 encoded by exon 1 of H19, both functioned as a positive regulator in osteogenic differentiation of human BMSCs in vitro and enhanced heterotopic bone formation in vivo. It was found that miR-675 directly downregulated TGF-β1 expression by targeting the 5′-UTR and coding sequence (CDS) region of *TGF-β1*, which subsequently inhibited phosphorylation of Smad3. LncRNA-H19/miR-675 also downregulated the mRNA and protein levels of histone deacetylase 4/5 (HDAC4/5), thereby decreasing the recruitment of these transcriptional repressors to the RUNX2-binding DNA sequence and enhancing the expression of osteogenic markers [110]. Another miRNA, miR-29b was found to positively regulate osteogenesis in MC3T3-E1 cells by downregulating TGF-β3 and other known anti-osteogenic factors, such as histone deacetylase 4 (HDAC4), activin A receptor type 2A (ACVR2A), Beta-catenin-interacting protein 1 (CTNNBIP1), and dual specificity phosphatase 2 (DUSP2) proteins, through direct binding to the 3’-UTR sequences of their transcripts, with potential contribution to osteoblast differentiation [111].

### 5.2. MicroRNAs and TGF-β Receptors (TGFBR)

TGF-β receptors are comprised of Type I receptor (TGFβRI or ALK5) and type II receptor (TGFβRII or Tgfbr2) [112]. TGF-β signalling is based on the coordination of the heteromeric complex of TGFBR1 and 2\ and specific intracellular Smad effector proteins [112]. Mice lacking either *TGFβRI/ALK5* or *TGFβRII* in MSCs resulted in a series of developmental abnormalities, such as defects in short, long bones and complications related to joint formation and growth plate maturation [113,114,115]. In addition, Maeda, Shingo et al. reported that the TGFβRI inhibitor SB431542 greatly enhanced osteogenic differentiation of mouse C2C12 cells by activating phosphorylation of BMP-specific R-Smads while suppressing I-Smads expression, as well as inducing the production of ALP, BSP, and matrix mineralization in human MSCs. Thereby, the findings of this study outlined a negative regulatory role of TGFβR1 in bone formation [116].

Let-7a-5p was upregulated in BMSCs of postmenopausal osteoporosis mice. It was demonstrated that the upregulated let-7a-5p could inhibit RUNX2 and Osx expression, directly target *TGFβRI*, and further inhibited osteogenic differentiation of BMSCs in these mice [117]. Contrastingly, let-7f-5p was upregulated during osteoblastic differentiation in rat BMSCs [118] but downregulated in the vertebrae of patients with glucocorticoid-induced osteoporosis [119]. The overexpression of let-7f-5p rescued the dexamethasone (Dex)-inhibited osteogenic differentiation of murine BMSCs in vitro and reversed the Dex-induced bone loss in vivo, through direct targeting of *TGFβRI* 3′-UTR, suggesting a positive role on osteogenesis [120]. Additionally, another positive regulator, the regulatory T cell derived-exosomal miR-142-3p promoted angiogenic effects in human umbilical vein endothelial cells (HUVECs), facilitated osteogenesis in BMSCs, and accelerated murine femoral fracture healing in vivo, by suppressing TGFBRI/Smad2 expression [121].

During endochondral and intramembranous ossification, the expression of different miR-181 isoforms, including miR-181a, miR-181b, and miR-181c was induced. The expression of miR-181a was increased in the course of the BMP-2- or BMP-6-induced C2C12 and MC3T3-E1 osteogenic stimulation in vitro and during mouse calvarial and tibial development in vivo. MiR-181a was strongly expressed during the later stage of osteogenic differentiation in primary calvarial osteoblasts and positively regulated osteoblastic differentiation through repressing the TGF-β signalling molecules, such as TGF-beta induced (*Tgfbi*) and *TGFβRI* by targeting their 3′-UTR [122].

A study conducted by Zhang, Xin et al. suggested that miR-140-5p is indispensable in regulating the mechanism of adipocyte/osteoblast differentiation through a C/EBP/miR-140-5p/TGFBR1 regulatory feedback loop. MiR-140-5p expression was increased in adipose tissue of db/db obese mice compared to their genetically-matched lean littermates. In addition, it was reported that the knockdown of miR-140-5p could enhance osteoblast differentiation, while its overexpression induced adipogenesis in stromal ST2 cells and preadipocyte 3T3-L1 via direct regulation of *TGFβRI* expression [123]. Hence, this finding support miR-140-5p as a potential target for new therapies aimed at managing metabolic disorders such as osteoporosis and obesity.

Furthermore, Zhang, Zheng et al. provided novel evidence that miR-223 expression could be induced in inflamed gingival tissues and was positively correlated with clinical parameters of periodontitis. MiR-223 acted as a negative regulator of osteogenic differentiation in periodontal ligament (PDL)-derived cells by targeting two important growth factor receptors genes: *TGFβR2* and fibroblast growth factor receptor 2 (*FGFR2*). This study also suggested that the downregulation of miR-223 was beneficial for PDL-derived cell osteogenesis by potentially mediating genes enriched in the MAPK signalling pathway, and thus, miR-223 is a potential biomarker for periodontitis [124].

Figure 3 and Table 1 summarize the emerging literature that supporting the involvement of miRNAs in regulation of TGF-β signalling, suggesting important regulatory roles in osteogenic differentiation.

## 6. MicroRNAs Regulating the Bone Morphogenic Proteins (BMP) Signalling Pathway

### 6.1. MicroRNAs and Bone Morphogenic Proteins (BMP) Ligands

Bone morphogenetic proteins (BMPs) comprise a group of over 30 proteins in humans and represent the largest subdivision of the TGF-β superfamily of ligands [126]. Among the BMPs, BMP-2, -4, -5, -6, -7, and -9 are thought to possess strongest osteogenic capacity [127]. The addition of BMP-2 was found to greatly increase OCN release [128] and bone formation in mouse experimental models [129,130]. Supplementation of BMP-7 facilitated osteogenic differentiation by elevating the expression of osteoblastic lineage-specific markers, ALP activity, and accelerating matrix mineralization [131,132]. BMP-9, a poorly characterized member of the BMP family is also shown to be one of the most potent osteogenic BMPs [133]. Conversely, deficiency of *BMP-2* and *BMP-4* resulted in a severe impairment of osteogenesis and osteoblast maturation [134]. Furthermore, *BMP*-7 knockout in mice caused neonatal death and resulted in skeletal patterning abnormalities confined to the rib cage, skull, and hindlimbs [135].

BMP-3 is mainly produced by osteoblasts and osteocytes in adult bone. BMP-3 was found to activate type IIB activin receptor (AcvrIIB)-Smad2/3 signalling to oppose the osteogenic activity of Smad1/5/8 and counteracted the responsiveness of other BMPs in osteoblast differentiation [136]. BMP-3 was also found to negatively regulate bone mass, as *BMP-3* disruption increased trabecular bone formation [136,137], whereas *BMP-3* overexpression limited skeletal progenitor cell differentiation into mature osteoblasts, by which both situations led to spontaneous fracture [136].

There is proven clinical applications for BMP-based treatments in patients, for example recombinant human BMP-2 and BMP-7, which have been approved for the treatment of open fractures of long bones, non-unions, and spinal fusion, with the comparable outcomes as with autologous bone graft [138]. The following sections will consider the effects of miRNAs on BMP regulation.

#### 6.1.1. MicroRNAs and BMP-2 Ligands

The expression of the miR-497~195 cluster, which belongs to the highly conserved miR-15 family, was strongly upregulated as age progressed during postnatal bone development and in the late differentiation stages of cultured primary mouse calvaria osteoblasts. A study by Grünhagen, Johannes et al. found that early expression of miR-195-5p strongly inhibited pre-osteoblast differentiation and mineralization through intracellular antagonization of BMP-responsive genes such as *Furin*, *Acvr2a*, *Bmpr1a*, *Dies1*, *Tgfbr3*, *Smad5*, *Ski*, *Zfp423*, *Mapk3*, and *Smurf1*. Overexpression of miR-195-5p was also found to interfere with the gene regulatory program of osteoblast differentiation in primary mice calvaria osteoblasts and MC3T3-E1 subclone 4 (MC4) cells. In addition, treatment with high-dose rhBMP-2 was found to overcome the inhibitory effect of miR-195 on early-stage osteoblast differentiation and mineralization. Together these findings indicate that the miR-497~195 miRNA cluster is important for regulating osteoblastic lineage shift from young, proliferative osteoblasts towards the mature, differentiated osteoblasts and osteocytes in adults [139].

Expression of miR-20a was significantly increased during human MSC osteogenic differentiation. Ectopic expression of miR-20a promoted osteogenic differentiation of human MSCs by upregulating BMP/RUNX2 signalling, and co-targeting 3′-UTR of peroxisome proliferator-activated receptor gamma (*PPARγ*), BMP and Activin membrane-bound inhibitor homolog (*Bambi*), and cysteine rich transmembrane BMP regulator 1 (*Crim1*) [140]. *PPARγ* is a suppressor of osteoblast phenotype and negative regulator of BMP/RUNX2 osteogenic signalling [141,142], whereas Bambi or Crim1 are the critical antagonists of the BMP signalling pathway [90]. In addition, miR-20a-5p positively contributed to osteogenic differentiation of human dental pulp stem cells (hDPSCs) in vitro and promoted regeneration of calvarial defects in mice by functionally targeting *Bambi* to activate the phosphorylation of Smad5 and p38 [143]. Therefore, the above functional studies identified miR-20a as positive regulator of osteogenesis by targeting the antagonists of BMP signalling pathway.

In addition to its involvement in regulating the TGF-β and TGFBR expression as mentioned in section above, miR-140-5p was also commonly enriched in undifferentiated human MSCs and was found to functionally inhibit osteogenic lineage commitment of MSCs by directly targeting *BMP-2* to maintain cell proliferation and MSC stemness [108]. The expression level of miR-204 was decreased in a time-dependent manner during osteogenic differentiation of rat BMSCs. The ectopic expression of miR-204 was found to downregulate the expression of RUNX2 and ALP and reduce the osteogenic capacity of rat BMSCs through direct targeting interaction with *BMP-2* mRNA [144].

Zhang, Guo-Ping et al. reported that expression of miR-98 via miRNA mimic transfection lowered osteogenic differentiation of human BMSCs by targeting *BMP-2*, while the miR-98 inhibitor group produced the opposite effect [145]. Recently, Zheng, Feng et al. found that inhibition of miR-98-5p enhanced cell viability and promoted cell proliferation and differentiation capacity in a high glucose-induced pre-osteoblastic model of diabetes mellitus associated osteoporosis. In addition, silencing of miR-98-5p prevented its direct targeting effect on *BMP-2*, sequentially resulting in activation of the PI3K/AKT/GSK3β signalling pathway and enhancement of osteoblast differentiation [146].

MiR-214 was investigated in an OVX rat model which underwent subsequent osteoporotic fracture operation. Silencing of miR-214 via antagomiR administration resulted in significantly increased bone mineral density and accelerated fracture healing in callus tissues. MiR-214 silencing was postulated to enhance fracture healing by upregulation of BMP-2 and Smad4 [147]. A study by Zhang, Hao et al. showed that, as compared with the healthy groups, high levels of miR-410 and low levels of BMP-2 were detected in serum and CD^14+^ peripheral blood mononuclear cells (i.e., PBMCs; precursors of osteoclasts) obtained from postmenopausal osteoporosis patients and the OVX mouse model. Dual luciferase reporter assays and Western blot also revealed a direct interaction between miR-410 and its target *BMP-2*, which may underly the pathogenesis of postmenopausal osteoporosis [148]. Taken together, the upregulation of miR-98, miR-214, and miR-410 may contribute to bone loss and deleterious effects on osteoblast proliferation and function.

Ankylosing spondylitis (AS, OMIM#106300) is a long-term inflammatory spinal arthritis which causes small bones in vertebrae to fuse with increased fracture susceptibility in the spinal column [149]. MiR-214 was found to impact fibroblast osteogenesis by targeting *BMP-2* and inactivating the BMP/TGF-β axis, presenting a potential treatment target for stimulating new bone formation in ankylosing spondylitis therapeutic [150].

MiR-93-5p was reported by Zhang, Ying et al. to facilitate proliferation while suppressing osteogenic differentiation of human BMSCs, which was accompanied by significantly reduced ALP activity and calcium nodule formation as well as the expression of BMP-2, RUNX2, and Osx [151]. Trauma-induced osteonecrosis of the femoral head (TIONFH) is a major complication that arises from femoral neck fractures, dislocation of the hip, and other hip trauma [152]. Increased miR-93-5p expression in TIONFH patients inhibited osteoblastic differentiation by reducing BMP-2 mRNA and protein expression via 3′-UTR targeting, which was accompanied by significant TIONFH bone necrosis with hip collapse and dysfunction. Hence, miR-93-5p may be a potential therapeutic target for the prevention of TIONFH.

MC3T3-E1, a mouse pre-osteogenic cell line that differentiates to osteoblast under the stimulation of BMP-2, is a well-known model for studying osteoblast differentiation [153]. The expression of miR-543 was significantly reduced during osteogenic differentiation in this model. It was also found that overexpression of miR-543 inhibited the proliferation and osteogenic differentiation of MC3T3-E1 cells via direct inhibition of BMP-2 [154]. Upregulation of another miRNA, miR-142, was found to repress cell viability while promoting apoptosis in MC3T3-E1 cells, thereby inhibiting osteoblasts differentiation. *BMP-2* was also confirmed as a target gene of miR-142. In addition, miR-142 was found to substantially reduce the p-Smad1/5/Smad1 and p-Smad1/5/Smad5 ratios, suggesting a suppressive role [155]. Li, Xuesen et al. identified for the first time that miR-342-5p was downregulated during osteoblast differentiation of MC3T3-E1 cells. It was found that aberrant expression of miR-342-5p inhibited proliferation, induced apoptosis, suppressed cell migration, and inhibited osteogenic differentiation of MC3T3-E1 or human MSCs, by functionally targeting both collagen type IV alpha 6 chain (*COL4A6*) and *BMP-2*. Silencing of miR-342-5p promoted cell proliferation, migration, and osteoblast differentiation via upregulating BMP-7, along with activation of the mitogen-activated protein kinase/ERK kinase (MEK)/extracellular-signal-regulated kinase (ERK) cascade [156].

LncRNA-KCNQ1OT1 was found to positively influence osteogenic differentiation of human BMSCs by sponging miR-214 as a competing endogenous RNA (ceRNA). MiR-214 is believed to target *BMP-2* and co-transfection of miR-214 inhibitor reversed the downregulation of *BMP-2*, *RUNX2*, *OPN*, and *OCN* genes, and the suppression of osteogenic differentiation that arose from the lncRNA-KCNQ1OT1 silencing. Additionally, miR-214 inhibitor transfection was found to overcome the reduction of p-Smad1/5/8, RUNX2, and Osx protein levels caused by the lncRNA-KCNQ1OT1 depletion [157]. Furthermore, lncRNA-MSC-AS1 also worked as a competing endogenous miRNA that sponged miR-140-5p activity as a positive regulator of BMP-2/Smad signalling. Thus, this promoted the osteogenic differentiation of mouse BMSCs and alleviated the progression of osteoporosis [158]. Additionally, lncRNA-Rhno1 was able to counteract the negative effects of miR-6979-5p on osteoblast differentiation of MC3T3-E1 cells by serving as a ceRNA for miR-6979-5p. The local administration of lncRNA-Rhno1 was able to enhance fracture healing in vivo, whereas the administration of miR-6979-5p agomiR into fracture sites worsened fracture healing by the direct targeting of *BMP-2* [159].

#### 6.1.2. MicroRNAs and Other Members of BMP Ligands

There is clear evidence that miRNAs regulate osteoblastic differentiation by targeting the BMP ligands other than those of BMP-2. Primary osteoblasts and MSCs isolated from miR-451a knockout mice showed increased osteogenic potential. Knockout of miR-451a could also relieve osteoporotic symptoms and promoted bone formation in the OVX mice model. This effect of miR-451a was attributed to targeting BMP-6; deletion of miR-451a resulted in upregulation of BMP-6, BMP receptor, and Smad1/5/8 signalling [160]. In addition, overexpression of miR-765 impaired the osteogenic differentiation of human BMSCs via targeting *BMP-6* and further reduced Smad1/5/9 phosphorylation upon osteogenic differentiation [161]. Additionally, *BMP-6* was shown to be a common direct target gene of miR-146a-5p/miR-146b-5p, as verified through the dual-luciferase reporter assay. Highly expressed lncRNA-ZNF710-AS1 was found to competitively sponge miR-146a-5p/miR-146b-5p, which was thought to enhance the ability of PDLSCs to differentiate into osteoblasts [162].

MiR-542-3p was shown to be downregulated during medicarpin-induced osteogenesis of mice calvarial osteoblastic cells. The 3′-UTR of *BMP-7* was confirmed as the putative targeting region of miR-542-3p. Interestingly, the overexpression of miR-542-3p suppressed osteogenic differentiation via the inhibition of the BMP-7/Smad-dependent pathway, whereas it inhibited cell proliferation and promoted osteoblast apoptosis through the inhibition of the BMP-7-mediated PI3K/survivin non-Smad signalling pathway. The silencing of miR-542-3p led to enhanced bone formation and bone strength and improved trabecular microarchitecture in OVX mice in vivo [163]. LncRNA-SNHG16 expression was markedly downregulated in human BMSCs harvested from osteoporosis patients, while miR-485-5p expression was markedly upregulated. Overexpression of lncRNA-SNHG16 was found to promote osteogenic differentiation of human BMSCs through positive regulation of BMP-7 expression and inhibition of miR-485-5p by acting as a ceRNA [164].

MiR-181a-3p was found to inhibit osteogenic differentiation of human BMCSs by targeting *BMP-10*, accompanied by downregulation of ALP activity and reduced expression of RUNX2 and activin receptor-like kinase (ALK), a type of BMP type I receptor [165].

BMP-3 is the most abundant BMP member in bone, which functions as an inhibitor of the osteogenic BMP family by activating the TGF-β/activin pathway [138]. Hence, miRNAs targeting *BMP-3* generally have positive regulatory effects on osteogenic differentiation. Overexpression of miR-450b was found to positively regulate the osteogenic differentiation of human adipose-derived mesenchymal stem cells (hADSCs) into osteoblasts in vitro as well as promote ectopic bone formation in vivo by targeting *BMP-3* [166].

A recent study by Sun, Jijiu et al. indicated the importance of lncRNA-SNHG3 in mediating osteolytic bone metastases in breast cancer, a condition develops when metastatic cancer cells break down too much of the bone, disrupting the balance between osteogenesis and osteoclast resorption in favour of the latter [167,168,169]. LncRNA-SNHG3 was reported to be dramatically upregulated in breast cancer cells. Knockout of lncRNA-SNHG3 resulted in reduced proliferation and migratory capacity of breast cancer cells and promoted osteogenesis in human BMSCs. Additionally, knockout of lncRNA-SNHG3 promoted bone regeneration in vivo via upregulating exosomal miR-1273g-3p expression and downregulating the expression of *BMP-3*, the verified direct target of miR-1273g-3p. LncRNA-SNHG3 knockdown also augmented osteoprotegerin (OPG) expression and further impeded osteoclast differentiation. Together these findings suggest that the SNHG3/miR-1273g-3p/BMP-3 axis may represent a novel target for the treatment of breast cancer bone metastases [167].

### 6.2. MicroRNAs and Bone Morphogenic Protein Receptor (BMPR)

There are three type I receptors of BMPs, comprising BMPR1A (ALK3), BMPR1B (ALK6), and AcvR1 (ALK2); and three type II receptors comprising BMPR2B, ACTR2A, and ACTR2B [170]. Overexpression of the C-terminal truncated BMP type II receptor (BMPR2), and deletion of the BMP type IA receptor (*BMPR1A*) in mouse osteoblasts were found to result in irregular calcification and low bone mass [171,172].

Overexpression of miR-100 [173] and miR-153 [174] have been shown to suppress osteogenic differentiation of human adipocyte-derived MSCs (ADSCs) and bone marrow-derived MSCs (BMSCs) by targeting BMP receptor type II (*BMPR2*). In addition, miR-155 expression was found to increase during the early-stage of BMP-9-induced osteogenic differentiation of MSCs, with a subsequent decreased in the later stage. It was also demonstrated that overexpression of miR-155 attenuated BMP-9-induced ALP and calcium deposition, and impaired osteoblastic lineage commitment of C2C12 cells and mouse embryonic fibroblasts (MEF) by directly suppressing both *RUNX2* and *BMPR2* translation. Subsequently, miR-155 overexpression was found to negatively regulate BMP-9-induced osteogenic differentiation by diminishing the protein expression of p-Smad1/5/8. In vivo, miR-155 agomiR administration inhibited ectopic bone formation [175]. MiR-494 was revealed to be elevated in C2C12 cells cultured under clinorotation conditions and in osteoblasts isolated from tail-suspended rats, simulating the microgravity unloading conditions. MiR-494 overexpression inhibited BMP-2-induced osteogenic differentiation of C2C12 cells by directly targeting both *RUNX2* and *BMPR2* [176].

MiR-1187 expression was strongly downregulated in medicarpin-treated osteoblast cultures. Overexpression of miR-1187 was found to inhibit osteoblast differentiation in primary mouse calvarial osteoblasts. Osteoblastic cells transfected with anti-miR-1187 vs. miR-1187 mimic showed more robust BMP-2-induced osteoblast differentiation, ALP activity, and mineral nodule formation. In addition, human osteoblastic cells transfected with miR-1187 mimic displayed reduced osteoblast differentiation and decreased in expression of osteogenic markers, in a manner similar to that of mouse osteoblasts [177]. Bioinformatics analysis and experimental validation via luciferase 3′-UTR reporter assay revealed both *BMPR2* and Cdc42 guanine nucleotide exchange factor 9 (*ArhGEF-9*) as putative target genes of miR-1187. Overexpression of miR-1187 targeted the components of BMP signalling and actin polymerization pathways, such as BMPR2, p-PAK, p-LIMK1, and p-cofilin. Additionally, by reducing actin polymerization and cortical protrusions formation, increased miR-1187 expression was found to decrease cell migration [177]. It was further revealed that miR-1187 negatively regulated osteogenic events by targeting BMP/BMPR2 signalling interaction, which, in turn, suppressed ArhGEF9-mediated Cdc42 activation and prevented F-actin stabilization. The silencing of miR-1187 in neonatal mice alleviated its inhibitory effects on actin cytoskeletal reorganization. Importantly, treatment of anti-miR-1187 in OVX mice was found to markedly improve trabecular bone microarchitecture in vivo [177].

MiR-195-5p expression was downregulated in the periodontal tissues of the orthodontic tooth movement (OTM) mouse model and in primary human periodontal ligament cells (PDLCs) cultured under cyclic tension strain. These models were used to mimic orthodontic mechanical loading of the tooth and alveolar bone remodelling, respectively. Negative correlation between miR-195-5p expression and osteogenic differentiation in these models was mediated by miR-195-5p directly targeting bone morphogenetic protein receptor-1A (*BMPR1A*), along with Wnt family member 3A (*Wnt3a*) and fibroblast growth factor 2 (*FGF2*) [178]. In addition, miR-23a was markedly elevated in periodontal ligament stem cells (PDLSCs) and gingival crevicular fluid of chronic periodontitis and gum disease patients. MiR-23a also acted as a negative regulator of osteogenesis in PDLSCs by targeting bone morphogenetic protein receptor-1B (*BMPR1B*) which further led to inhibition of Smad1/5/9 phosphorylation [179]. In addition, miR-125b was under expressed during osteogenic differentiation of human BMSCs. Increased expression of miR-125b resulted in inhibition of BMSC osteogenic differentiation by targeting *BMPR1B*, while its miR-125b knockdown resulted in enhanced segmental bone defect repair capacity [180].

Recent research has investigated the role of methyltransferase-like 3 (METTL3) in progression of osteoporosis. First, METTL3 was found to be downregulated in osteoporosis. Further research indicated that METTL3 mediated m6A methylation of LINC00657 in osteogenesis progression. The role if METTL3 in osteoporosis was attributable to its function as a ceRNA that upregulated *BMPR1B* via sponging miR-144-3p [181]. Figure 4 and Table 2 described miRNAs with important contribution to the regulation of bone forming osteoblasts through their effects on integral targets of the BMP ligands/BMPR signalling.

## 7. MicroRNA Regulation of the Smad Cascades

### 7.1. MicroRNAs and the BMP-Regulated Smads (R-Smads)—Smad1 and Smad5

The R-Smads, known as Smad1, 5, and 8 are the downstream transcription mediators of BMP signalling, which can be phosphorylated and activated by BMP type I receptors [80,183]. The BMP-Smad1/5/8 signal transduction pathway is essential for the regulation of osteoblast proliferation and differentiation [93,112]. After the assembling of a complex between the phosphorylated Smad1/5/8 and Co-Smad, Smad4, the complex is then translocated to the nucleus for activation of transcription factor RUNX2/Cbfa1 [184,185]. RUNX2/Cbfa1 serves as a master transcription factor to regulate the expression of major bone matrix protein genes, such as *ALP*, *OCN*, *OPN*, *ON*, *BSP*, and collagen type I alpha 1 chain (*COL1A1*), which eventually stimulates mineral deposition and mineralized bone nodule formation [186,187]. Osteoblast-specific *Smad1* conditional knockout mice resulted in impairment of osteoblast proliferation and differentiation, development of osteopenia, and partial disruption of BMP signalling [188]. Combined depletion of *Smad1* and *Smad5* resulted in severe chondrodysplasia. Therefore, both Smad1 and Smad5 are crucial in mediating BMP signal for endochondral bone formation [188,189,190]. The contribution of Smad8 to skeleton development are thought to be less than those of Smad1 and Smad5 [93].

#### 7.1.1. MicroRNAs and Smad1

MiR-26a showed very low expression in undifferentiated human ADSCs, but expression increased under differentiation, and peaking at hADSC terminal differentiation [191]. MiR-26a expression was also increased following osteogenic differentiation of BMSCs [192]. MiR-26a negatively modulated human ADSC terminal differentiation into osteoblastic lineage by targeting *Smad1* [191,192], while it positively regulated Wnt signalling to promote the osteogenic differentiation of BMSCs [192]. Collectively, these findings indicate dual function of miR-26a in promoting BMSC osteogenic differentiation but inhibiting ADSC osteogenic differentiation.

Members of the miR-30 family, such as miR-30a, miR-30b, miR-30c, and miR-30d, were downregulated during osteogenic differentiation stimulated by various stimuli, including stimulation of BMP-2 in C2C12 and MC3T3E1 cells [193,194], and with mixed enamel matrix proteins, Emdogain^®^ for biomineralization or osteogenic induction in MC3T3E1 cells [195,196]. MiR-30s inhibited osteoblast differentiation of MC3T3-E1 by targeting *Smad1* and *RUNX2*, further leading to decreased expression of the p-Smad1/5 protein in miR-30 overexpressing cells [194]. Therefore, miR-30s contributed to the early-stage of osteoblast differentiation by inhibition of Smad1 and RUNX2. The miRNA miR-100 also specifically targeted *Smad1* to inhibit the BMP-2-induced osteogenic differentiation of MC3T-3E1 cells and primary mouse BMSCs, resulting in a decrease in Smad1 protein levels without affecting other members of the Smad pathway, including Smad3 and Smad4, Smad7, and RUNX2 [197].

Diabetes is often accompanied by abnormal bone metabolism and disorders of calcium and phosphorus metabolism. Diabetes can interfere with bone formation, increase the risk of fractures, and prevent bone fracture healing. The jaws of patients with diabetes mellitus (DM) often show loss of alveolar bone associated with osteoporosis, sometimes leading to loosening or even loss of teeth [198,199]. Dental implants are an important method to restore missing teeth. However, skeletal complications and the abnormal metabolic environment caused by DM lead to pathological changes, which greatly reduce the success rate of dental implants [200].

MiR-203-3p was identified to be upregulated in the mandibles of type 2 diabetes (T2DM) rats. It was found that miR-203-3p inhibition of osteogenesis in this animal model and in rat BMSCs cultured in a high glucose medium was through inhibition of BMP/Smad pathway by targeting of *Smad1* mRNA. These findings suggest that miR-203a-3p can serve as a possible therapeutic target diabetic osteoporosis, fracture healing, tooth stability, and implant osseointegration [201].

#### 7.1.2. MicroRNAs and Smad5

MiR-222-3p acted as a negative regulator of osteogenic differentiation in human foetal MSCs by functionally targeting both *Smad5* and *RUNX2*. Inhibition of miR-222-3p activity significantly enhanced mRNA and protein levels of Smad5 and RUNX2, and promoted osteogenic effects by activation of the Smad1/5/8 signalling pathway [202]. MiR-128-3p was enriched in exosomes isolated from aged rat BMSCs. Exosomal miR-128-3p was confirmed to directly target and negatively modulate the expression of *Smad5*. In vitro transfection of miR-128-3p mimic in young rat BMSCs decreased osteogenic differentiation, while transfection of miR-128-3p inhibitor accelerated osteogenic differentiation in aged rat BMSCs. Additionally, inhibition of miR-128-3p improved better femoral fracture healing in rats in vivo. In short, miR-128-3p suppressed BMSC osteogenic differentiation, and these negative effects could be circumvented by Smad5 overexpression [203].

During the process of osteogenic induction in human BMSCs, lncRNA-KCNQ1OT1 was significantly upregulated while miR-320a expression was markedly downregulated. LncRNA-KCNQ1OT1 promoted OCN, OPN, and RUNX2 expression by acting as a ceRNA that sponged miR-320a, upon osteogenic differentiation. MiR-320a was also found to directly inhibit *Smad5*, thereby negatively regulating osteogenic differentiation [204].

Studies have shown that miR-133 and miR-135 inhibited BMP-2-induced C2C12 osteogenic differentiation by downregulating RUNX2 and p-Smad1/5/Smad5, respectively [193]. Consistent with this, miR-133a, miR-133b, and miR-135 were downregulated in human dental pulp stem cells (hDPSCs) grown on dental implant titanium surfaces and showed a negative correlation with osteogenesis and expression of RUNX2 and Smad5 [205]. MiR-21 [206] and miR-24-3p [207] were markedly decreased in osteogenic-differentiated human periodontal ligament stem cells (hPDLSCs), and the overexpression of both miRNAs suppressed osteogenic differentiation in these cells by targeting *Smad5*, resulting in subsequent downregulation of RUNX2.

Other studies also indicated that the levels of miR-106b-5p/miR-17-5p [208] and miR-155 [209] were downregulated during the process of osteogenic differentiation in BMP-2-treated C2C12 and MC3T3-E1 cells. Overexpression of miR-106b-5p/miR-17-5p and miR-155 inhibited osteoblast differentiation of BMP-2-treated osteoblastic C2C12 and MC3T3-E1 cells in vitro likely through direct targeting of *Smad5* mRNA. In addition, silencing of miR-106b-5p/miR-17-5p promoted bone formation in OVX mice through Smad5 regulation [208]. Furthermore, ectopic expression of miR-155 inhibited mRNA and protein expression of both native and phosphorylated Smad5 [209].

Cyclic stretch stress (12%) was shown to inhibit the osteogenic potential of MC3T3-E1 cells, and miR-132-3p expression was also reported to be activated under this condition. Overexpression of miR-132-3p was found to negative modulate osteogenic differentiation via decreased expression of p-Smad5 and Smad5 in MC3T3-E1 cells under cyclic stretch [210].

Ellur, Govindraj and colleagues investigated the effect of a protein-rich maternal diet on foetal bone mass in offspring. It was found that modal mice fed with a high-protein diet delivered offspring with significantly reduced body weight and length, decreased new-born skeletal mineralization and ossification, and compromised bone microarchitecture throughout life. This effect was due to a decline in osteoblast cell maturation. A small RNA sequencing study found that miR-24-1-5p was highly upregulated in the osteoblasts from the high-protein diet group. Target prediction and validation studies confirmed that miR-24-1-5p targeted 3′-UTR of *Smad5*, and inhibited osteoblasts mineralization possibly through inhibition of BMP-2-induced osteogenesis. Contrastingly, inhibition of miR-24-1-5p in cultured mouse calvarial osteoblasts derived from new-born offspring of the high protein group was found to rescue the suppressive effects of miR-24-1-5p on osteoblast maturation and Smad5 expression [211].

### 7.2. MicroRNAs and the TGF-β-Regulated Smads (R-Smads)—Smad2 and Smad3

The R-Smads Smad2 and 3 are activated by TGF-β receptors and regulate TGF-β-mediated signalling in response to osteoblast proliferation and differentiation [80,183]. The complex formed from phosphorylated Smad2/3 and Co-Smad, Smad4 result in activation or repression of downstream genes in the nucleus [80,183]. Smad2/3 was found to inhibit *RUNX2/Cbfa1* expression in murine experimental models [212,213]. Activated Smad3 recruited class II histone deacetylases 4 and 5 (HDAC4/5) to mediate the TGF-β-induced transcriptional repression of *RUNX2/Cbfa1* function in differentiating osteoblasts [213].

#### 7.2.1. MicroRNAs and Smad2

Extracellular vesicles (EVs) derived from neural EGFL-Like 1 (NELL1)-modified mesenchymal stem cells exhibited stronger osteo-inductive ability in rat BMSC owing to the downregulation miR-25-5p. MiR-25–5p suppressed osteogenic differentiation by targeting *Smad2* and further compromising the activation of SMAD and extracellular signal-related kinase 1 and 2 (ERK1/2) pathway. In addition, it was demonstrated that the 3D hydrogel system assembled with *Nell1*/EVs is beneficial for acellular bone regeneration on calvarial defect rat model, evidenced with better bone formation in the animals. Therefore, *Nell1*/EVs is a promising strategy for bone defect healing with favourable osteogenic capability through regulation of miR-25-5p/Smad2 signalling [214].

MiR-10b expression was positively associated with bone formation marker genes, such as *ALP*, *RUNX2*, and *OPN*, and negatively correlated with adipogenic markers including CCAAT/enhancer-binding protein alpha (*CEBPα*), peroxisome proliferator-activated receptor gamma (*PPARγ*), and adipocyte fatty acid-binding protein 2 (*aP2*) in clinical samples obtained from osteoporosis patients. Overexpression of miR-10b enhanced osteogenic differentiation and inhibited adipogenic differentiation of human ADSCs in vitro. Implantation of miR-10b overexpressing human ADSC xenografts in mice resulted in enhanced ectopic bone formation in vivo. *Smad2* was identified as a putative target of miR-10b, and miR-10b was responsible for regulating osteogenic differentiation and lineage commitment of human ADSCs partly via regulating the TGF-β signalling pathway [215]. In addition, increased expression of miR-214-5p was observed in Dex-induced adipogenic differentiation of human BMSCs as compared to the undifferentiated cells. Furthermore, miR-214-5p overexpression favoured adipogenic differentiation, and, in turn, suppressed the expression of osteogenic markers *ALP*, *RUNX2*, *OCN*, and *COL1A1* in human BMSCs. The upregulation of miR-214-5p also decreased TGF-β, p-Smad2, and collagen type IV α1 chain (COL4A1) protein expression in human BMSCs [216]. Another miRNA miR-6315, was found to promote osteogenic differentiation and to alleviate methotrexate (MTX)-induced increased adipogenesis [217]. MTX is an antimetabolite-based cancer chemotherapeutic agent proven to induce bone loss and bone marrow adiposity [218] It was also shown that the effects of miR-6315 involvement in osteogenesis/adipogenesis regulation is potentially via inhibition of TGF-β/Smad2 signalling. MiR-6315 might be applied as a potential therapeutic target in treatment of long-term skeletal side effects from intensive chemotherapy [217,218].

TGF-β signalling is crucial for skeletal development. BMP and its receptors are generally thought to induce early chondrogenesis and to stimulate differentiation of mesenchymal cells into osteoblasts, while TGF-β ligand and its receptors regulate chondrocytes proliferation and differentiation [219]. Skeletal stem cells isolated from the epiphyseal region of the human foetal femur showed increased expression of genes associated with chondrogenic differentiation, such as SOX9 and collagen type II (Col II), while cells derived from the diaphyseal region of the femur exhibited increased expression of genes correlated with osteoblast phenotype, including RUNX2, ALP, collagen type I (Col I), and ON. MiR-146a was found to be prominently expressed by diaphyseal cells of the human foetal femur compared to epiphyseal cells. Putative target and functional analysis confirmed *Smad2* and *Smad3* as the direct target genes of miR-146a in foetal femur cells. Transient overexpression of miR-146a in the epiphyseal cells resulted in downregulation of *Smad2* and *Smad3* translation and upregulation of the osteogenic *RUNX*, in addition to downregulation of the chondrogenic *SOX9* [220]. The same study indicated that TGF-β3 was capable of stimulating cell differentiation into hypertrophic chondrocytes, which was accompanied by upregulation of type X collagen (*Col X*) mRNA expression [219]. Overexpression of miR-146a was found to prevent chondrocyte hypertrophic differentiation and reduced TGF-β3-induced Col X upregulation, potentially by attenuating the TGF-β3 ligand signal. Together, these findings indicate that miR-146a has a positive effect on osteogenesis and a negative effect on chondrogenesis [220].

#### 7.2.2. MicroRNAs and Smad3

Osteonecrosis (ON) is a condition characterized by the disruption of the blood supply to bone [221]. Non-traumatic osteonecrosis of the femoral head (ONFH) is caused by elevated intra-osseous pressure and abnormal formation of bone marrow fat content, leadings to ischemia and the death of osteocyte and bone marrow cells [221]. Treatment procedures, including core decompression and joint replacement surgery, are generally applied to improve the course of ONFH [222]. However, these traditional treatment strategies can only reduce the intramedullary pressure to a moderate degree, are expensive, and associated with only weak clinical benefits [223].

MiR-708 [224], miR-181d [225], and miR-596 [226] expression was found to be upregulated, while Smad3 expression was found to be downregulated in the bone marrow samples of patients with steroid-induced ONFH. Functional analysis revealed that overexpression of miR-708, miR-181d, and miR-596 suppressed osteogenic differentiation and proliferation of steroid-induced BMSCs by reduction of mRNA and protein levels of Smad3, which disrupted Smad3 and RUNX2 interaction and inactivated TGF-β1 signalling. Interestingly, *Smad3* was proven as the direct target of all three miRNAs by Western blots and luciferase reporter assays. In addition, the level of miR-423-5p in the serum of ONFH patients was significantly increased, which was positively correlated with femoral head collapse incidents and negatively correlated with the level of adiponectin [227]. MiR-423-5p was also found to be upregulated in the bone marrow samples of ONFH patients and suppressed osteoblastic differentiation and cell viability of human BMSCs by targeting *Smad3* in non-traumatic osteonecrosis [228]. Together, these findings support the involvement of miR-708, miR-181d, miR-596, and miR-423-5p in ONFH progression.

Trauma, including bone fractures, can elicit translocation of lipopolysaccharide (LPS) endotoxins from the gut, which potently trigger inflammatory responses, and clinical conditions including endotoxinemia/sepsis, which pose challenges to fracture healing [229]. LPS has been identified as a major bacterial mediator of bone resorption [230]. Aside from inducing hypertrophy and immature callus, LPS was also found to reduce bone mineral density and strength in a rat model of fracture healing [229]. Liu, Hongzhi et al. reported that LPS inhibited osteogenic differentiation of pre-osteoblast MC3T3-E1 cells by upregulating Smad3 and downregulating miR-23b. Moreover, overexpression of miR-23b protected against LPS-induced inhibition of BMP-2-stimulated osteogenic differentiation by targeting *Smad3* [231].

Expression of miR-300 was markedly reduced upon treatment with the osteoanabolic bioactive peptide PepC in rat osteoblasts. Ectopic expression of miR-300 negatively regulated osteoblast differentiation and matrix mineralization by targeting *Smad3* and disruption of cross-talk among Smad3, β-catenin, and RUNX2 in rat calvarial osteoblast cultures. *Smad3* was confirmed as the direct target of miR-300 during osteogenic differentiation, and downregulation of Smad3 destabilized β-catenin, thereby inhibiting its translocation to the nucleus. In vivo silencing of miR-300 in neonatal rat pups and adult female OVX rats abolished the inhibitory action of miR-300 on osteoblast differentiation and on Smad3/β-catenin/RUNX2 cross-talk. Micro-CT analysis showed improved trabecular microarchitecture and bone formation following miR-300 silencing in the OVX rat model. In addition to these findings from experimental models, increased miR-300 expression was observed in serum from 30 osteoporotic patients [232].

### 7.3. MicroRNAs and the Partner or Common Smad (Co-Smad)—Smad4

Smad4 is another crucial component of the TGF-β/BMP signalling pathway, in which it participates as a Co-Smad through forming complexes with other R-Smad members, Smad2/3 and Smad1/5/8 [80,183]. Expression of Smad4 is required for the maintenance of osteoblast differentiation in MSCs [93,112]. The conditional knockdown of *Smad4* in pre-mature osteoblasts resulted in abnormalities in collagen matrix, in bone collagen-modifying enzyme gene expression, impairment of mineralizing response, and development of osteogenesis imperfecta-like features characterized by severe growth retardation and spontaneous fracture [233].

The expression of miR-224 was significantly increased in three types of mesenchymal stem cells; adipose-derived mesenchymal stem cells (MSC-A), bone marrow-derived mesenchymal stem cells (MSC-B), and umbilical cord-derived mesenchymal stem cells (MSC-U), compared with human osteoblasts. This finding suggests that miR-224 is important for maintaining the stemness of MSCs. The expression of miR-224 gradually decreased during MSC osteogenic differentiation, and the knockdown of miR-224 stimulated osteoblast differentiation in vitro. The activity of miR-224 in this study was thought to be via negative regulation of *Smad4*. Overexpression of miR-224 also inhibited osteoblast differentiation by downregulating osteogenic-associated markers, including OCN, OPN, BSP, and RUNX2, in addition to the suppression of other related pathways; signal transducer and activator of transcription 3 (STAT3), Akt, nuclear factor NF-kappa-B p50 subunit (NF-κB p50), ERK1/2, and p38 MAPK [234].

MiR-224-5p was upregulated in glucocorticoid-treated rat BMSCs. Functional analysis showed that miR-224-5p could serve as a reciprocal regulator that suppressed osteogenesis but promoted adipogenic differentiation in rat BMSCs [235]. *Smad4* was identified as functional target of miR-224-5p, and the Smad4-Taz axis was identified as the intrinsic regulatory pathway responsible for mediating adipo-osteogenic differentiation in rat BMSCs. Park, Jin Seok et al. found that direct binding of Smad4 to the transcriptional coactivator Taz facilitated nuclear translocation of Taz and promote osteogenic differentiation of MSCs through the formation of the nuclear Taz-RUNX2 complex [236]. The inhibition of miR-224-5p was shown to reduce the incidence and severity of SONFH in vivo [235].

MiR-1323 was upregulated in human specimens of atrophic non-union fracture, a condition characterized by atrophy and arrested healing at the site of bone fracture, without radiological evidence of callus formation. MiR-1323 upregulation was accompanied by downregulation of BMP-4 and Smad4, along with other osteogenic-related markers, such as ALP, Col I, and RUNX2. The overexpression of miR-1323 inhibited osteogenic differentiation in mesenchymal stromal cells by targeting of *BMP-4* and *Smad4*, thereby suppressing protein expression. The silencing of miR-1323 also promoted nuclear translocation of the transcriptional coactivator Taz through the activation of BMP-4/Smad4 activity, thus supporting osteogenesis. Furthermore, the silencing of miR-1323 facilitated fracture healing and osteogenesis in a femoral fracture model, supporting the potential for miR-1323 antagomiR therapy in the treatment of atrophic non-union fracture [182].

In TNF-*α*-treated mouse BMSCs, a model of systemic inflammation, increased expression of miR-146a was observed in addition to decreased expression of Smad4. Upregulation of miR-146a was also found to promote the TNF-α-mediated inhibition of osteogenesis in vitro and in vivo by direct inhibition of *Smad4* [237].

Another miRNA, miR-144-3p, was also found to negatively regulate osteogenic differentiation. MiR-144-3p was found to be upregulated in murine C3H10T1/2 mesenchymal cells, in which it inhibited proliferation in the G0/G1 phases of the cell cycle. MiR-144-3p was shown to exert its inhibitory role in osteoblastic proliferation and differentiation through targeted suppression of *Smad4* [238]. In addition, miR-664-3p expression was markedly downregulated during the osteogenic differentiation of the C3H10T1/2 mesenchymal cells and in pre-osteoblast MC3T3-E1 cells. Upregulation of intracellular miR-664-3p via miRNA mimic transfection resulted in inhibition of osteoblast activity and matrix mineralization in vitro. Osteoblastic miR-664-3p transgenic mice exhibited an osteoporotic bone phenotype with concomitant reduction in bone strength and bone mass, due to suppression of osteoblast function. Bioinformatic miRNA-target prediction and experimental validation confirmed that miR-664-3p directly attenuated *Smad4* and *Osx* expression in both mice and humans. Furthermore, the circulating level of miR-664-3p was increased in osteoporosis patients relative to individuals without osteoporosis, and specific inhibition of miR-664-3p by subperiosteal injection with miR-664-3p antagomiR could partially rescue against ovariectomy-induced bone loss in mice [239].

### 7.4. MicroRNAs and the Inhibitory Smads (I-Smads)—Smad6 and Smad7

The inhibitory Smads (I-Smads), Smad6 and Smad7, are involved in the negative regulation of TGF-β signalling. I-Smads competing with Smad1/5/8 and Smad2/3 for binding to the activated type I receptors and for interaction with Smad4 [240]. While Smad6 selectively limits BMP signalling [241], Smad7 can inhibit both BMP and TGF-β signalling [242]. Additionally, I-Smads were found to antagonize TGF-β/BMP signalling by recruiting Smad specific E3 ubiquitin protein ligase 1 and 2 (Smurf1 and Smurf2), to induce proteasome-mediated degradation of R-Smads or type I receptors [243,244,245,246]. *Smad7* mutants showed impaired ability to recruit Smurf2, which compromised their inhibitory activity [243]. Therefore, knockout of the I-Smads has potential to facilitate the activation of R-Smads and their translocation into the nucleus. As I-Smads are inhibitors of osteogenic differentiation, targeted inhibition of I-Smads by miRNAs results in positive modulatory effects on osteogenesis.

Jia, Jie et al. found that miR-17-5p expression was markedly reduced in non-traumatic osteonecrosis samples compared with osteoarthritis control samples. BMP-2-induced overexpression of miR-17-5p was shown to directly downregulate *Smad7*, which promoted nuclear translocation of β-catenin, enhanced expression of RUNX2 and COL1A1, which together facilitated proliferation and differentiation in the HMSC-bm stromal cell line [247]. In addition, increased lncRNA-HOTAIR was detected in non-traumatic osteonecrosis samples, which inhibited the expression miR-17-5p and, in turn, increased the expression of its target gene *Smad7* [248]. A separate study conducted by Feng, Xiaobo et al. and Han, Ning et al. revealed that the circular RNAs circHGFA and circ_0058792 suppressed cell proliferation and osteogenic differentiation of osteoblast lineage cells in SONFH via inhibition of miR-25-3p and miR-181-5p binding to *Smad7* [249,250]. Another study by Fang, San Hong et al. also demonstrated that the miR-15b, which was upregulated in BMSCs, markedly enhanced osteogenic differentiation capacity and thereby relieved SONFH by targeting *Smad7*, while promoting TGF-β-associated p-Smad2/3 protein expression [251].

A large and growing body of literature has investigated the vital regulatory role of miR-21 in osteoblast/osteogenic differentiation, and its contributions to bone cellular physiology and pathology in various bone-related disorders. The upregulation of circ-PVT1 exerted protective effects against SONFH by modulating the miR-21-5p-controlled Smad7/TGF-β signalling pathway [252]. In addition, MSC-derived exosomes extracted from osteoporosis patients was believed to suppress osteogenesis via the miR-21/Smad7 regulatory axis [253].

Osteogenic differentiation of human BMSCs into osteoblasts achieved by pulsed electromagnetic field (PEMF) stimulation was shown to induce the expression of miR-21, which activate the TGF-β signalling pathway. PEMF activated cell proliferation and differentiation and induced miR-21 expression in BMSCs of younger female patients to a greater extent than BMSCs from older women. In the same study, PEMF-induced osteogenic differentiation of BMSCs was accompanied by decreased Smad7 expression, in addition to enhanced expression of both TGF-β2-activated Smad2 and *RUNX2* [254].

Another recent study observed upregulation of miR-21-5p, as well as the pro-osteogenic miR-129-5p and miR-378-5p in human MSCs with differentiation induced through treatment with serum obtained from runners after a half marathon. Concomitant reduced expression was observed in miR-21-5p’s target genes: *PTEN* and *Smad7*, along with increased protein levels of Akt/p-Akt and Smad4. In addition, a consequent upregulation of RUNX2 expression was also observed upon miR-21-5p-mediated osteogenic differentiation. Together, these findings suggest that miR-21-5p, miR-129-5p, and miR-378-5p regulate MSC osteogenic differentiation in response to physical exercise [255].

Expression of miR-21 was also upregulated during the osteogenic differentiation in another in vitro study, in which mouse BMSCs were isolated from bone cavities. BMSCs isolated from miR-21 knockout mice showed decreases in osteogenic differentiation and new bone formation compared to the wild-type mice. It was further found that miR-21 regulated bone formation in BMSCs, in part through the Smad7-Smad1/5/8-RUNX2 pathway [256]. MiR-21 was also upregulated during osteogenic differentiation in murine multilineage cells (MMCs) induced by BMP-9, and miR-21 overexpression was found to sustain the activation of BMP-9-Smad1/5/8 signalling by directly inhibiting *Smad7* [257]. Additionally, syringic acid treatment in mouse MSCs [258] and phytol treatment in C3H10T1/2 cells [259] were found to stimulate osteogenic differentiation through the miR-21-mediated downregulation of *Smad7*, which, in turn, activated RUNX2 expression.

Another study elucidated further mechanistic insights into miR-21 regulation of Smad7, finding in MC3T3-E1 cells that stimulation of osteogenic differentiation and mineralization by miR-21 was by the inhibition of *Smad7* expression through translational inhibition rather than by mRNA decay [260]. MiRNA profiling in MC3T3-E1 cells under hypoxia found ten upregulated miRNAs, miR-21-5p, miR-252a-5p, miR-9b-5p, miR-122-3p, miR-7a-5p, miR-200-5p, miR-181a-5p, miR-128a-5p, miR-210-5p, and miR-145-5p, and seven downregulated miRNAs, miR-30b-3p, miR-146a-5p, miR-182-5p, miR-522-3p, miR-34a-5p, miR-204-5p, and miR-184b-5p. The expression of Smad3 and Smad7 decreased after hypoxia, but there was no significant difference in the expression of Smad2 or Smad4 between MC3T3E1 cells under hypoxic vs. normoxic conditions [261]. Three highly homologous matching sites were also noted between the miRNA-21-5p and the 3′-UTR of *Smad7.* Sustained hypoxia inhibits osteogenic differentiation through direct downregulation of RUNX2 expression [262]. Elevated miR-21-5p expression rescued hypoxia-induced RUNX2 inhibition by upregulating RUNX2 protein expression in a Smad7-dependent manner [261].

A study performed by Chen, Yinan et al. found that the overexpression of miR-590-5p, which shares the exact same seed sequence as miR-21-5p, promoted osteoblast growth and differentiation via activation of TGF-β signalling through direct inhibition of *Smad7* expression in high glucose-treated MC3T3-E1 cells. In addition, in MC3T3-E1 cells the increased expression of osteogenesis-related genes, p-Smad2 and p-Smad3 promoted by miR-590-5p overexpression was markedly inhibited by the transfection of *Smad7*-overexpressing plasmid [263].

Vishal et al. identified a significant increase in miR-590-5p expression during osteoblast differentiation of mouse MSCs and human BMSCs obtained from women aged 29, 30, and 36 years. In addition, transient transfection of MG-63 osteoblasts with miR-590-5p mimic resulted in elevation of both calcium deposition and expression of osteoblast differentiation marker genes. The study further identified that miR-590-5p promoted osteoblast differentiation by indirectly stabilizing the RUNX2 protein against Smad specific E3 ubiquitin protein ligase 2 (Smurf2)-mediated ubiquitination and proteasomal degradation, via directly suppressing *Smad7* expression [264]. MiR-590 was also found to induce proliferation and extracellular matrix depositions in human umbilical cord mesenchymal stem cells (hUMSCs) by reducing the mRNA and protein levels of Smad7, implicating this miRNA as an important regulator of stem cell growth, tissue repair, and wound healing [265].

A number of other miRNAs than those described above are reported to have involvement in the regulation of Smad7. He, Guisong et al. found that miR-877-3p overexpression enhanced osteoid formation via Smad7 inhibition in MC3T3-E1 cells with osteoblastic differentiation induced by TGF-β1 [266]. In addition, miR-2861 induced the osteogenic differentiation of rat BMSCs through direct interaction with *Smad7* [267]. Similarly, miRNA-324-3p was found to facilitate icariin-induced MC3T3-E1 osteoblastic differentiation and cell proliferation via *Smad7* inhibition [268]. Furthermore, the natural product zingerone stimulated osteoblast differentiation of human BMSCs by upregulating miR-200c, which, in turn, targeted *Smad7* and enhanced the expression of ALP, OCN, Osx, and RUNX2 [269].

While many studies highlight the role of I-Smad7 in osteogenic differentiation, there is a relatively small body of literature reporting the involvement of I-Smad6 in this function. MiR-20a, a mechanosensitive miRNA was found to enhance fluid shear stress-mediated osteogenic differentiation of rat BMSCs via the activation of BMP-2 signalling pathway by targeting *Bambi* and *Smad6* [270]. In addition, silencing of hsa-circ-0107593 was found to upregulate miR-20a-5p expression, in which stimulated osteogenic differentiation of human ADSCs through targeting of *Smad6* [271].

Figure 5 and Table 3 detail miRNAs with observed regulatory activity in the Smad signalling cascades and with relevance to osteogenesis or bone-related disease.

## 8. Conclusions

We have provided a detailed overview of the roles of miRNAs as key modulators of osteogenic differentiation, bone development, homeostasis, and repair processes, with particular focus on the canonical TGF-β/BMP signalling cascades. A number of miRNAs are shown to negatively regulate osteogenic differentiation by targeting transcripts from genes coding for the extracellular ligands and receptors of the TGF-β/BMP and Wnt signalling, as well as the transcription factor β-catenin, TCF, RUNX2, and Osx. On the other hand, some miRNAs positively regulate osteogenesis by directly targeting negative osteogenic regulators, such as the inhibitory Smad7, GSK-3β, APC, DKK1, SOST, sFRPs, and WIF-1. Additionally, certain miRNAs appear to possess capacity for both displaying both positive and negative regulation of these key signalling pathways.

The significance of miRNA-mediated regulation of osteogenic differentiation, skeletal development, and homeostasis is now apparent. The major focus of future research in this area will be (a) further mechanistic insights into miRNA-mediated regulation of bone homeostasis and (b) translation of these insights into the clinic. Bioinformatic studies have rapidly accelerated research in this area, with a vast number of miRNAs implicated in osteoblastic TGF-β/BMP signalling from predominant approaches including microarray analysis and small RNA-sequencing. Owing mainly to the vast number of miRNAs with predicted roles in bone homeostasis, there still exists a major gap between these predictions from high-throughput or in silico approaches and in vitro/in vivo validation of mechanistic function, including the confirmation of target genes of these small RNAs. A significant challenge to such advancements is the lack of consensus on the most suitable model for studying bone homeostasis, in part due to the vast complexity of bone physiology in vivo. Here, we presented findings from myriad in vitro model systems, which highlights the careful consideration required when interpreting the miRNA research findings. Primary cells more closely represent model in vivo cellular behaviour but present challenges, including obtaining sufficient bone fragments for cell culturing, finite lifespan, and phenotype instability in long-term culture [272]. Osteosarcoma cell lines ease some of these experimental challenges due to unlimited growth capacity and phenotypic stability but may not fully resemble the behaviour of primary osteoblast cells [272]. This lack of consensus extends beyond choice of cell type, and careful consideration must be exercised when selecting the most appropriate in vitro or in vivo model that most accurately represents the particular osteoblast function of interest.

In spite of these challenges and considerations, significant progress has been made in various clinical applications of miRNAs involved in osteoblast functions as potential diagnostic indicators and therapeutic targets for the treatment of bone disorders. TAmiRNA, a European biotech company specialized in miRNA diagnostics, has developed various diagnostic solutions from their licensed miRNAs, including as biomarkers in osteoporosis. Their lead product, the OsteomiR™ miRNA biomarker assay, enabled the accurate assessment of the risk of a first fracture in female patients with postmenopausal osteoporosis and type 2 diabetes (T2DM) [273]. However, there are still many limitations and barriers to the transition of miRNA-based diagnostic biomarkers and therapeutics. For example, MRX34 (ClinicalTrials.gov Identifier: NCT01829971), the first miRNA (miR-34a) mimic to reach a phase 1 clinical study for the treatment of primary liver cancer and other malignancies, was terminated due to the detection of five immune-related serious adverse events [273]. The major potential issues faced by miRNA-based therapy include (a) the biological barriers and miRNA-induced immune response, (b) problems of delivery, (c) cytotoxicity effects, (d) potential off-targets and non-specific tissue distribution, (e) the low stability of miRNA-based oligonucleotides in vivo, and (f) degradation by RNases and rapid renal excretion [274]. In the last decades, certain measures, such as chemically modified miRNA mimics and anti-miRNA oligonucleotides, and novel delivery systems have been developed to increase the efficiency and efficacy of miRNA therapeutics [275].

In conclusion, a large body of evidence shows that miRNAs play major roles in the osteogenic process and the development of bone tissue. MiRNA studies are not only ushering in a new era of basic bone biology research but are also advancing the development of new diagnostic and treatment strategies for a variety of bone diseases in clinical practice.

## Figures and Tables

**Figure 1 ijms-24-06423-f001:**
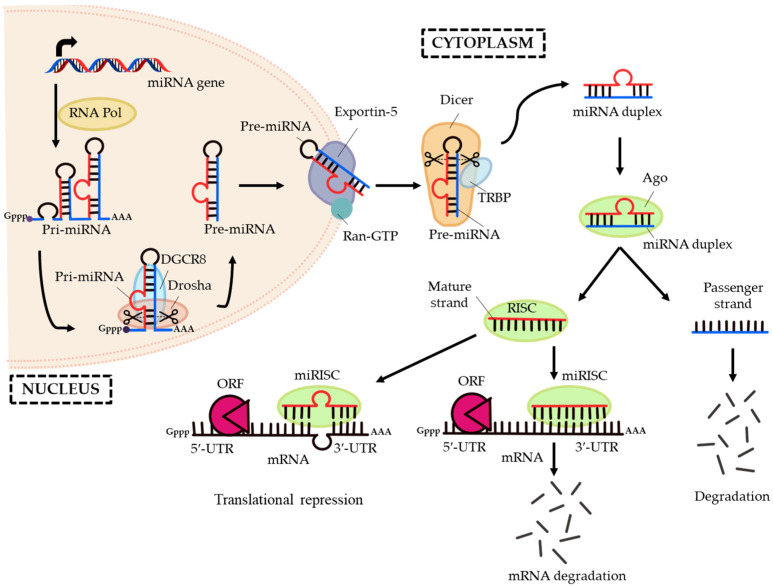
The pathway of microRNA biogenesis and gene silencing.

**Figure 2 ijms-24-06423-f002:**
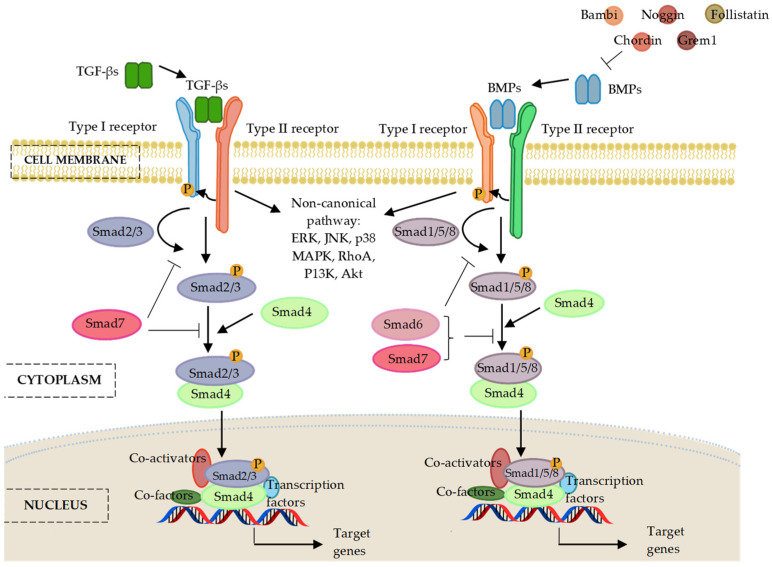
Transforming growth factor-beta (TGF-β)/bone morphogenic protein (BMP) signalling pathway.

**Figure 3 ijms-24-06423-f003:**
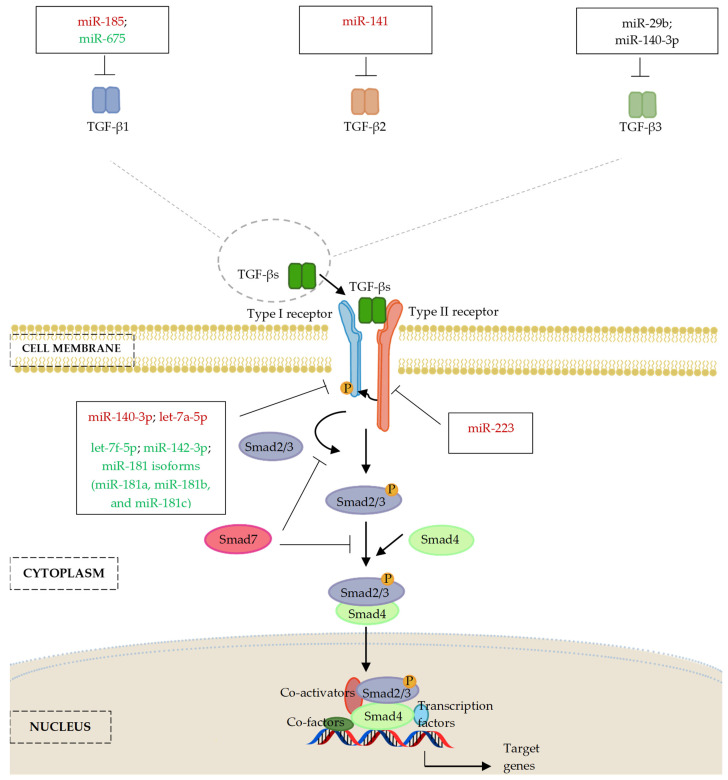
Schematic representation of microRNA-mediated regulation of the transforming growth factor-β (TGF-β) signalling pathway. **Notes:** MicroRNAs in green implied positive regulation on osteogenesis, and microRNAs in red implied negative regulation on osteogenesis, while microRNAs in black text indicate both positive and negative effects on osteogenesis.

**Figure 4 ijms-24-06423-f004:**
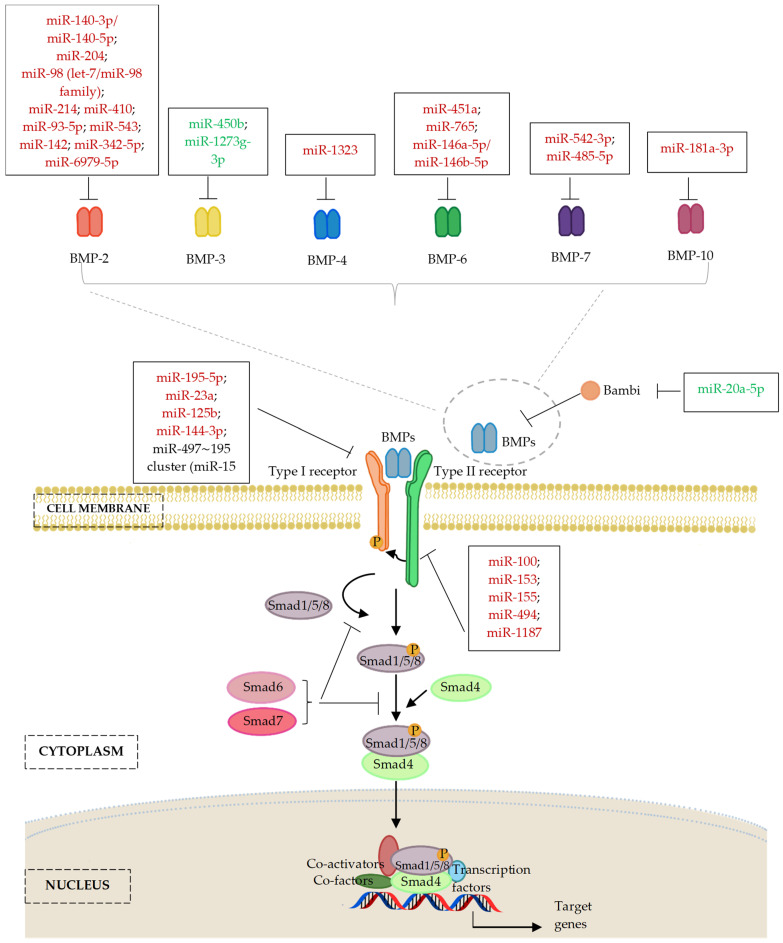
MicroRNAs with regulatory effects on components of bone morphogenic protein (BMP) signalling. **Notes:** MicroRNAs in green indicate positive regulation on osteogenesis, and microRNAs in red indicate negative regulation on osteogenesis, while microRNAs in black text indicate both positive and negative effects on osteogenesis.

**Figure 5 ijms-24-06423-f005:**
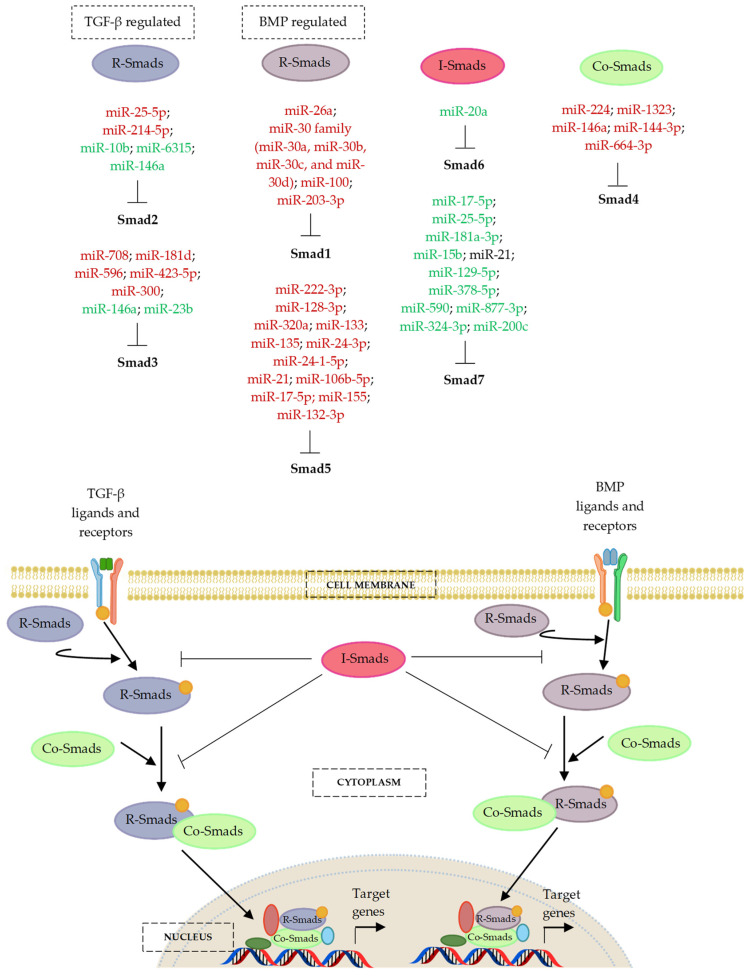
MicroRNAs regulating components of the Smad signalling pathway. **Notes:** MicroRNAs in green indicate positive regulation on osteogenesis, and microRNAs in red indicate negative regulation on osteogenesis.

**Table 1 ijms-24-06423-t001:** List of microRNAs implicated in the TGF-β signalling pathway in various model systems and cell types.

miRNAs	Protein Encoded by Direct Target Gene(s)	Model Systems and Cell Types	Gene Targeting Effect of miRNAs on Osteogenesis	References
* **MicroRNAs and Transforming Growth Factor-β (TGF-β) Ligands** *				
** MiR-185 **	TGF-β1	Bone and peripheral blood samples obtained from patients with spinal cord injuries induced by thoracolumbar spine compression fractures	−	[103]
	TGF-β1	Bone and peripheral blood samples obtained from male and female patients of ankle fracture (23–60 years old); hFOB1.19 cells	−	[104]
MiR-675	TGF-β1	Primary human MSCs; osteogenically induced human MSCs implanted subcutaneously in BALB/c homozygous nude (nu/nu) mice (5 weeks old)	+	[110]
MiR-141	TGF-β2	Male and female SONFH patients (mean age: 60.45 years) and control patients with femoral neck fractures (mean age: 60.85 years); SONFH male and female Sprague–Dawley (SD) rat model (body weight 100 ± 20 g)	−	[107]
MiR-29b	TGF-β3, HDAC4, ACVR2A, CTNNBIP1, and DUSP2; COL1A1, COL5A3, COL4A2	MC3T3-E1 clone 4 cells; primary rat calvaria osteoblast cells	−/+	[111]
** MiR-140-3p **	TGF-β3	MC3T3-E1 cells	−/+	[109]
* **MicroRNAs and TGF-β Receptors (TGFBR)** *				
** MiR-140-5p **	TGFβRI	db/db obese mice and their genetically matched lean littermates (10 weeks old); BMSCs isolated from femurs and tibias of C57 mice (4 weeks old); stromal ST2 and preadipocyte 3T3-L1 cells	−	[123,125]
** Let-7a-5p **	TGFβRI	BMSCs derived from OVX female C57BL/6J mice (4 weeks old, body weight 20 ± 2 g); human embryonic kidney HEK cells	−	[117]
** Let-7f-5p **	TGFβRI	BMSCs derived bone marrow collected from long bones (tibias and femurs) of mice (8 weeks old); glucocorticoid-induced osteoporosis model of male C57BL/6 mice (8 weeks old)	+	[120]
** MiR-142-3p **	TGFβRI	Male C57BL/6 mice fracture model and traumatic brain injury model (5–6 weeks old); regulatory T cells culture; HUVECs; mouse BMSCs isolated from fresh femoral bone marrow samples; isolation and treatment of TregD-Exos	+	[121]
**MiR-181 isoforms** (**miR-181a**, **miR-181b**, and **miR-181c**)	TGFβRI, Tgfbi	C2C12 cells; MC3T3-E1 cells; primary calvaria osteoblasts; tibia and calvaria samples harvested from male new-born (24 h) C57BL/6 wildtype mice (1–6 weeks old)	+	[122]
MiR-223	TGFBRII, FGFR2	Periodontitis patients and healthy controls; periodontal ligament tissues collected from 30 teeth, which were premolars and third molars extracted from healthy donors (14–23 years old)	−	[124]

**Notes:** The microRNAs in **bold in red** are those validated to regulate multiple target genes in the same signalling pathways or in different signalling discussed across this review; the age, weight, and sex of human and animal subjects involved in the studies are indicated where this information was available in the original manuscripts; the “+” sign indicates positive regulation on osteogenesis, while the “−” sign indicates negative regulation on osteogenesis.

**Table 2 ijms-24-06423-t002:** List of microRNAs implicated in the BMP signalling pathway in various model systems and cell types.

miRNAs	Protein Encoded by Direct Target Gene(s)	Model Systems and Cell Types	Gene Targeting Effect of miRNAs on Osteogenesis	References
* **MicroRNAs and Bone Morphogenic Protein (BMP) Ligands** *				
** MiR-20a **	PPARγ, Bambi, Crim1	Human BMSCs from healthy donors	+	[140]
** MiR-20a-5p **	Bambi	Human DPSCs isolated from premolars collected from donors who underwent teeth extraction Due to orthodontic needs (12–14 years old); female nude BALB/c mice with calvarial defects (32 months old)	+	[143]
**MiR-140-5p** and **miR-140-3p**	BMP-2	Human MSCs	−	[108]
** MiR-140-5p **	BMP-2	Mouse BMSCs	−	[158]
MiR-204	BMP-2	BMSCs isolated from male Sprague–Dawley (SD) rats (4 weeks old)	−	[144]
** MiR-98 (let-7/miR-98 ** ** family) **	BMP-2	BMSCs isolated from femoral head of male and female patients undergoing hip arthroplasty (mean age: 59.1 ± 6.2 years)	−	[145]
** MiR-98-5p **	BMP-2	High glucose-treated MC3T3-E1 cells	−	[146,139]
** MiR-214 **	BMP-2	OVX female Sprague–Dawley (SD) rats (12 weeks old, body weight 250–300 g) which underwent subsequent osteoporotic fracture operation	−	[147]
	BMP-2	Fibroblasts isolated from the cut hip joint capsule ligament of male and female patients with AS Involving both hips and requiring joint replacement (average age 40.7 ± 2.4 years), and control patients with femoral neck fracture (free of AS and other immune diseases) who needed open surgery or joint replacement (average age 40.7 ± 2.4 years)	−	[150]
	BMP-2	Human BMSCs	−	[157]
MiR-410	BMP-2	Female patients with postmenopausal osteoporosis (age range 50–59 years old, mean age: 55.6 ± 4.8 years) and healthy female subjects (age range 50–59 years old, mean age: 55.1 ± 4.6 years); female C57BL/6 mice (5 weeks old, weight between 18–22 g); CD^14+^ peripheral blood mononuclear cells (PBMCs)	−	[148]
MiR-93-5p	BMP-2	Male and female patients with femoral neck fractures; human BMSCs isolated from the bone marrow of donors with trauma	−	[151]
MiR-543	BMP-2	MC3T3-E1 cells	−	[154]
** MiR-142 **	BMP-2	MC3T3-E1 subclone 14 cells	−	[155]
MiR-342-5p	COL4A6, BMP-2	MC3T3-E1 cells, human MSCs procured from bone marrow aspirate and hMSC-TERT cell line; male and female fracture patients including hand fracture (25–56 years old), intra-articular calcaneal fracture (24–59 years old), healthy controls (24–59 years old)	−	[156]
MiR-6979-5p	BMP-2	Male C57BL/6J mice fracture modal (6 weeks old); MC3T3-E1 cells	−	[159]
** MiR-1323 **	BMP-4	Human male atrophic non-union fracture specimens and standard healing fracture specimens collected during open reduction/internal fixation (ORIF); MSCs; male Sprague–Dawley (SD) rat model of femoral fracture (13–14 weeks old, body weight 400–500 g)	−	[182]
MiR-451a	BMP-6	Bone samples of wild-type and miR-451a-knockout (KO) mice with or without OVX (6 weeks old), and miR-451a overexpression mice following OVX; primary osteoblasts isolated from neonatal mice (3 days old); mouse BMSCs obtained from femurs and tibias	−	[160]
MiR-765	BMP-6	Human MSCs	−	[161]
**MiR-146a-5p** and **miR-146b-5p**	BMP-6	Human PDLSCs isolated from normal impacted third molars collected from human subjects (16–35 years old)	−	[162]
MiR-542-3p	BMP-7	Mouse calvarial osteoblasts; OVX female BALB/c mice (6 weeks old)	−	[163]
MiR-485-5p	BMP-7	Bone marrow samples and primary human BMSCs harvested from osteoporosis patients and healthy controls	−	[164]
** MiR-181a-3p **	BMP-10	Human BMSCs obtained from healthy volunteers	−	[165]
MiR-450b	BMP-3	Human adipose tissue obtained from donors (20–45 years old) undergoing liposuction; scaffolds loaded with human ADSCs infected with lenti-450b implanted subcutaneously into the upper dorsal surface of male NOD/SCID mice (6 weeks old)	+	[166]
MiR-1273g-3p	BMP-3	Human breast cancer cells, MCF-7, HS598t, and MDA-MB-231; human normal mammary epithelial cells, MCF-10A; co-culture of shSNHG3 transfected MD-MB-231 with human BMSCs; SNHG3 loss-of-function analysis in female C57BL/6 mice which underwent segmental defect operation of femur (2 months old)	+	[167]
* **MicroRNAs and Bone Morphogenic Protein Receptor (BMPR)** *				
** MiR-100 **	BMPR2	Human ADSCs isolated from adipose tissues obtained from patients undergoing tumescent liposuction	−	[173]
MiR-153	BMPR2	Human BMSCs isolated from bone marrow samples of young subjects (<30 years old) and older subjects (>60 years old) with slight or severe osteoporosis	−	[174]
** MiR-155 **	BMPR2	HEK293 human embryonic kidney cell lines; human C2C12 myoblast cell lines; mouse embryonic fibroblasts (MEF); ectopic in vivo bone formation assay of transfected MEF cells in athymic female nude mice (4–6 weeks old)	−	[175]
MiR-494	BMPR2	C2C12 and HEK293T cells; osteoblasts isolated from tail-suspended rats mimics a simulated microgravity environment	−	[176]
MiR-1187	BMPR2, ArhGEF-9	Primary calvarial osteoblasts harvested from mouse pups (1–2 days old); miR-1187 gain-of-function and loss-of-function analysis in BALB/c neonatal pups (1–2 days old) and post-OVX female BALB/c mice (6 weeks old)	−	[177]
** MiR-195-5p **	BMPR1A, Wnt3a, FGF2	Human PDL tissues obtained from premolars that previously extracted from healthy individuals (14–20 years old) for orthodontic purposes; PDL cells subjected to cyclic tension stress (CTS) for modelling of orthodontic mechanical loading in vitro; C57BL/6 mouse model of mechanical loading tooth movement	−	[178]
MiR-23a	BMPR1B	Patients with moderate to advanced periodontitis (18–60 years old) and healthy subjects; gingival crevicular fluid samples; human PDLSCs isolated from tissues in the middle of teeth roots	−	[179]
MiR-125b	BMPR1B	Human BMSCs isolated from the posterior iliac crest of young healthy male volunteers; male BALB/c nude mice model of bone defects (7 weeks old, body weight 16–23.5 g)	−	[180]
**MiR-497∼195 cluster** (**miR-15 family**)	BMP-responsive genes, such as *Furin*, *Acvr2a*, *Bmpr1a*, *Dies1*, *Tgfbr3*, *Smad5*, *Ski*, *Zfp423*, *Mapk3*, and *Smurf1*	Bone tissue samples harvested from C57BL/6 wild-type mice; primary calvaria osteoblasts harvested from new-born mice (P0-P4); MC3T3-E1 subclone 4 cells	−/+	[139]

**Notes:** The microRNAs in **bold in red** are those confirmed to regulate multiple target genes in the same signalling pathways or in different signalling discussed across this review; the age, weight, and sex of human and animal subjects involved in various microRNA studies are indicated, where this information was provided in original manuscripts; the “+” sign indicates positive regulation on osteogenesis, while the “−” sign indicates negative regulation on osteogenesis.

**Table 3 ijms-24-06423-t003:** List of microRNAs implicated in Smad signalling cascades in various model systems and cell types.

miRNAs	Protein Encoded by Direct Target Gene(s)	Model Systems and Cell Types	Gene Targeting Effect of miRNAs on Osteogenesis	References
* **MicroRNAs and the BMP-Regulated Smads (R-Smads)—Smad1 and Smad5** *				
** MiR-26a **	Smad1	Human ADSCs isolated from adipose tissue obtained from the subcutaneous abdominal depot during herniotomy	−	[191]
	Smad1	C57BL/6J mice (4–6 weeks old); mouse BMSCs isolated from the bone marrow cavities of femur and tibia; mouse ADSCs isolated from scraps of subcutaneous adipose tissues	−	[192]
MiR-30 family (miR-30a, miR-30b, miR-30c, and miR-30d)	Smad1	MC3T3-E1 cells; mouse BMSCs prepared from the bone marrow of femur and tibia harvested from male C57B/L6 mice (2 months old)	−	[194]
** MiR-100 **	Smad1	MC3T3-E1 cells; mouse BMSCs isolated and harvested from femur and tibia bone marrow of male C57B/L6 mice (10 weeks old)	−	[197]
MiR-203-3p	Smad1	Diabetic male Sprague–Dawley (SD) rats (10 weeks old, body weight 290–310 g); high-glucose treated rat BMSCs isolated from mandible of male SD rats (8 weeks old) and C3H101/2 clone 8 cells	−	[201]
MiR-222-3p	Smad5	Human foetal MSCs	–	[202]
MiR-128-3p	Smad5	Male femoral fracture Sprague–Dawley (SD) rats (3 months old); BMSCs obtained from young (4 weeks old) and old (72 weeks old) male SD rats	−	[203]
MiR-320a	Smad5	Human BMSCs isolated from the bone marrow of healthy subjects	−	[204]
**MiR-133** and **miR-135**	Smad5	C2C12 cells	−	[193]
**MiR-133a**, **miR-133b**, and **miR-135**	Smad5	Human DPSCs cultured on titanium disks in vitro	−	[205]
** MiR-24-3p **	Smad5	PDLSCs isolated from human first premolars extracted for orthodontic reasons from donors (14–20 years old)	−	[207]
** MiR-24-1-5p **	Smad5	Female C57BL/6 mice (8–10 weeks old); mouse offspring; primary mouse calvarial osteoblasts obtained from new-born pups of the control and high-protein group	−	[211]
** MiR-21 **	Smad5	PDLSCs isolated from human first premolars extracted for orthodontic reasons from donors (10–14 years old); miR-21 gain-of-function and loss-of-function analysis of transfected PDLSCs implanted on immunocompromised beige mice (nu/nu nude mice) (10 weeks old)	−	[206]
MiR-106b-5p/**miR-17-5p**	Smad5	C2C12 cells; MC3T3-E1 cells; OVX female C57BL/6J mice (6 weeks old)	−	[208]
MiR-155	Smad5	MC3T3-E1 cells and HEK293 cells	−	[209]
MiR-132-3p	Smad5	MC3T3-E1 cells loaded with cyclic stress	−	[210]
* **MicroRNAs and the TGF-β-Regulated Smads (R-Smads)—Smad2 and Smad3** *				
** MiR-25-5p **	Smad2	BMSCs isolated from Sprague Dawley rats (4 weeks old) and infected with an adenovirus vector encoding the full-length sequence of *Rattus* *norvegicus Nell1*, followed by *Nell1*/EVs isolation; in vivo evaluation of bone repair with the *Nell1*/EV-hydrogel conducted on calvarial defect male SD rat model (6 weeks old)	−	[214]
MiR-10b	Smad2	Human ADSCs isolated from adipose tissue collected from healthy women who underwent Liposuction surgery; osteoporosis patients who had a fracture caused by falling without obvious violence; miR-10b gain-of-function and ectopic bone formation model of NOD/SCID mice (8 weeks old)	+	[215]
** MiR-214-5p **	TGF-β, Smad2, COL44A1	PTA-1058 human BMSC cell line	−	[216]
MiR-6315	Smad2	MC3T3-E1 pre-osteoblastic cells and 3T3 F442A pre-adipocytic cells	+	[217]
** MiR-146a **	Smad2, Smad3	Osteogenic diaphyseal and chondrogenic epiphyseal foetal femur cells population isolated from foetal femurs samples at 7–9 weeks post conception	+	[220]
MiR-708	Smad3	Bone marrow samples of patients with SONFH and patients with ONFH after a previous fracture of the femoral neck (both ranged 20–50 years old); human BMSCs isolated from the proximal end of femur after inserting the tapered awl into the femoral canal during THA; glucocorticoid-treated MSCs	−	[224]
** MiR-181d **	Smad3	Whole-bone marrow samples from the marrow cavity of patients undergoing total hip arthroplasty due to steroid-induced necrosis of the femoral head and control patients with secondary femoral head necrosis after the old femoral neck fracture (both ranged 20–50 years old); human BMSCs isolated from the proximal end of femur	−	[225]
MiR-596	Smad3	Bone marrow samples from patients with SONFH and from patients with femoral neck fracture who underwent total hip replacement (both ranged 25–50 years old); glucocorticoid-treated BMSCs	−	[226]
** MiR-423-5p **	Smad3	Serum samples from SONFH patients (50.5 ± 2.7 years old, 45% male) and healthy volunteers (49.5 ± 3.1 years old, 55% male)	−	[227]
	Smad3	Bone marrow samples from non-traumatic ONFH patients and OA control patients; human BMSCs and HEK293T cells	−	[228]
MiR-23b	Smad3	Lipopolysaccharides-treated MC3T3-E1 cells	+	[231]
MiR-300	Smad3	Primary rat osteoblast cells isolated from neonatal rat pups; human osteoblasts; femur and tibia collected from PepC-treated Wistar rats (3 months old); miR-300 gain-of-function and loss-of- function analysis in rat pups (2–3 days old); miR-300 gain-of-function and loss-of-function analysis in OVX Wistar albino rats (3 months old, body weight 180 g); blood serums from postmenopausal osteoporotic and non-osteoporotic human subjects	−	[232]
* **MicroRNAs and the Partner or Common Smad (Co-Smad)—Smad4** *				
** MiR-224 **	Smad4	MSC cell lines, including MSC derived from bone marrow (MSC-B), MSC umbilical cord-derived (MSC-U), MSC adipose-derived (MSC-A), and human skull osteoblasts (HCO); HEK293T cells	−	[234]
** MiR-224-5p **	Smad4	BMSCs extracted from the tibias and femurs of Sprague–Dawley (SD) rats (6 weeks old); SONFH male Sprague–Dawley SD rat model (12 weeks old)	−	[235]
** MiR-1323 **	Smad4	Human male atrophic non-union fracture specimens and standard healing fracture specimens collected during open reduction/internal fixation (ORIF); MSCs; male Sprague–Dawley (SD) rat model of femoral fracture (13–14 weeks old, body weight 400–450 g)	−	[182]
** MiR-146a **	Smad4	BMSCs isolated from male C57Bl6 mice (8-weeks old)	−	[237]
MiR-144-3p	Smad4	C3H10T1/2 cells	−	[238]
MiR-664-3p	Smad4, osterix	MC3T3-E1 cells; HEK293T cells; C3H10T1/2 cells; bone tissues of C57BL/6J mice (8 weeks old); conditional miR-664-3p transgenic C57BL/6J mice; miR-664-3p loss-of-function analysis in post- operated OVX female C57BL/6J mice (16 weeks old); peripheral blood samples of osteoporotic female patients with fracture and non-osteoporotic patients (50–89 years old)	−	[239]
* **MicroRNAs and the Inhibitory Smads (I-Smads)—Samd6 and Smad7** *				
** MiR-17-5p **	Smad7	Bone marrow specimens obtained from the proximal end of the femurs derived from non- traumatic ONFH patients and OA control patients; primary human BMSCs derived from the patients; commercial hMSC-BM cells	+	[247]
	Smad7	Bone marrow specimens obtained from the proximal end of the femurs derived from non- traumatic ONFH patients and OA control patients; primary human BMSCs derived from the patients; commercial hMSC-BM cells	+	[248]
** MiR-25-3p **	Smad7	Bone marrow specimens were harvested from patients with steroid-induced ONFH (25–68 years old, mean 50.0 years) and control patients with femoral neck fracture (33–68 years old, mean 54.6 years); human BMSCs	+	[249]
** MiR-181a-3p **	Smad7	BMSCs derived from patients with steroid-induced ONFH; methylprednisolone-induced ONFH Sprague–Dawley (SD) female rats (body weight 190–220 g); MC3T3-E1 cells	+	[250]
** MiR-15b **	Smad7	Bone marrow tissues collected from patients with glucocorticoid-induced osteonecrosis of the femoral head and patients with secondary ONFH receiving total hip replacement (both ranged 25–50 years old); human BMSCs; HEK293T cells	+	[251]
** MiR-21 **	Smad7	MSCs extracted from osteoporosis patients and healthy adults	−	[253]
	Smad7	BMSCs extracted from fresh human bone marrows of ‘young’ female patients (21–30 years old) and ‘older’ female patients (31–65 years old) which treated with PEMF	+	[254]
	Smad7	BMSCs isolated from the femur and tibia bone cavities of female C57BL/6J wild-type mice and miR-21-knockout (KO) mice (5 weeks old); wild-type and miR-21-KO mouse model of calvarial bone defects (body weight 20–22 g)	+	[256]
	Smad7	Human HEK293 cells, colorectal carcinoma HCT116 cells, and murine multilineage cells (MMCs): murine C2C12 myoblasts cell lines, and mouse embryonic fibroblasts (MEFs)	+	[257]
	Smad7	Syringic acid-treated mouse MSCs	+	[258]
	Smad7	MC3T3-E1 cells	+	[260]
**MiR-21-5p**, miR-129-5p, miR-378-5p	PTEN, Smad7	Serum samples collected before and immediately after a 21.1 km half marathon from male amateur runners (median age 40.2 ± 8 years); human BMSCs	+	[255]
** MiR-21-5p **	Smad7	SONFH male Sprague–Dawley (SD) rat model and control groups (10 weeks old); rat BMSCs isolated from bone marrow tissues removed from the proximal femur during total hip replacement	−	[252]
	Smad7	Hypoxia-treated MC3T3-E1 cells	+	[261]
** MiR-21a **	Smad7	Phytol-treated C3H10T1/2 cells	+	[259]
** MiR-590-5p **	Smad7	High glucose-treated MC3T3-E1 cells	+	[263]
** MiR-590-5p **	Smad7	Human BMSCs obtained from female subjects aged 27, 29, 30, and 36 years; mouse MSC C3H10T1/2 cells and human MG-63 cells	+	[264]
** MiR-590 **	Smad7	Human UMSCs	+	[265]
MiR-877-3p	Smad7	MC3T3-E1 cells; miR-877-3p gain-of-function and loss-of-function analysis in female NOD/SCID mice (4 weeks old, body weight 17–19 g)	+	[266]
MiR-324-3p	Smad7	Icarrin-treated MC3T3-E1 cells	+	[268]
MiR-200c	Smad7	Zingerone-treated human BMSCs	+	[269]
** MiR-20a **	Smad6, Bambi	Fluid sheer stress-treated MC3T3-E1 cells	+	[270]

**Notes:** The microRNAs in **bold in red** are those confirmed to regulate multiple target genes in the same signalling pathways or in different signalling discussed across this review; the age, weight, and sex of human and animal subjects involved in various microRNA studies are indicated, where this information was available in the original manuscripts; the “+” sign indicates positive regulation of osteogenesis, while the “−” sign indicates negative regulation of osteogenesis.

## Data Availability

Not applicable.

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
