# Peer review of "Post-Transcriptional Regulatory Crosstalk between MicroRNAs and Canonical TGF-β/BMP Signalling Cascades on Osteoblast Lineage: A Comprehensive Review"

_ijms, 2023, doi:10.3390/ijms24076423_

Round 1

Reviewer 1 Report

For the introduction, the authors should provide a schematic of the pathways/factors mostly involved in the osteogenesis process, as this will help the readers to go through the review.

The authors mentioned in paragraph 2 that microRNAs work in a concerted way alongside with other long non-coding RNAs but it will be relevant to describe also more in details ceRNAs as modulator of miRNA and mRNA functionality, defined also as miRNA sponges. 

In paragraph 3, it would be better to add a subparagraph on major diseases related to the alteration of the osteogenesis process, such as osteoporosis. It might be highlighted what are the limitations of the current therapy and why more studies on miRNAs will be beneficial for the treatment of the disease. 

For the paragraph 6, the authors should organize the subparagraphs on the role of the microRNAs on the control of the osteogenesis (miRNAs promoting osteogenesis vs miRNA with inhibitory effect on this process, and maybe a subparagraph on miRNAs displaying both positive and negative effect) rather than on the BMP factors targeted by miRNAs. I would suggest also to similarly organize the paragraph 7 focused on the TGFB-Smad signalling (positive /negative effect on differentiation).

Overall, the review is written in a good English but need to be revised for more concise sentences.

Reviewer 2 Report

Post-Transcriptional Regulatory Crosstalk between MicroRNAs and Canonical TGF-β/BMP Signalling Cascades on Osteoblast Lineage: A Comprehensive Review

Manuscript entitled “Post-Transcriptional Regulatory Crosstalk between MicroRNAs and Canonical TGF-β/BMP Signalling Cascades on Osteoblast Lineage: A Comprehensive Review” by Loh et al., is an interesting review article study. Here, the authors comprehensively reviewed and described the role of miRNA to manage the gene expression in osteogenic differentiation by regulating multiple signalling cascades and essential transcription factors, including the transforming growth factor-beta (TGF-β)/bone morphogenic protein (BMP), Wingless/Int-1(Wnt)/β-catenin, Notch, and Hedgehog signalling pathways. MiRNA interactions, importance in different signaling pathways as well as their mechanism of action for the development of possible therapeutics were explained.

Overall, the information presented in this review is useful for the researchers and I approve its publication after some minor updates. 

Minor comments: I suggest that these comments to be updated before publication.

§  The content provided in this review article is huge and very informative including the tables, but it will helpful for the readers if it have any flowchart or a figure to explain the interaction of various microRNAs with the specific pathways like TGF-β/BMP, Wingless/Int-1(Wnt)/β-catenin and Notch.

Reviewer 3 Report

The authors present a review for a general audience of the crosstalk between miRNAs and TGF-beta/BMP signaling in osteoblast related research. They also summarized the functional relevance of bone development related miRNA-target events and provided certain useful analysis guidelines for future studies in the field. The review is quite comprehensive and covers widely, I found the topics covered in this review very interesting for the scientific community, particularly for those researchers who study post-transcriptional regulatory mechanisms, and it will be useful as a summary and source of references.

The review is thorough and comprehensive, however, I think the authors should address one issue to make the review more clearer. Figure. 1 is unnecessary. Here (for table 1,2 & 3), I strongly recommend creating new signaling cascades pattern diagrams (even supplementary figures), summing up the miRNAs and their targets.
